# Conformal Off-Policy Prediction in Contextual Bandits

**Muhammad Faaiz Taufiq\***
Department of Statistics
University of Oxford

**Jean-Francois Ton\*†**
AI-Lab-Research
Bytedance AI Lab

**Rob Cornish**
Department of Statistics
University of Oxford

**Yee Whye Teh**
Department of Statistics
University of Oxford

**Arnaud Doucet**
Department of Statistics
University of Oxford

## Abstract

Most off-policy evaluation methods for contextual bandits have focused on the expected outcome of a policy, which is estimated via methods that at best provide only asymptotic guarantees. However, in many applications, the expectation may not be the best measure of performance as it does not capture the variability of the outcome. In addition, particularly in safety-critical settings, stronger guarantees than asymptotic correctness may be required. To address these limitations, we consider a novel application of conformal prediction to contextual bandits. Given data collected under a behavioral policy, we propose *conformal off-policy prediction* (COPP), which can output reliable predictive intervals for the outcome under a new target policy. We provide theoretical finite-sample guarantees without making any additional assumptions beyond the standard contextual bandit setup, and empirically demonstrate the utility of COPP compared with existing methods on synthetic and real-world data.

## 1 Introduction

Before deploying a decision-making policy to production, it is usually important to understand the plausible range of outcomes that it may produce. However, due to resource or ethical constraints, it is often not possible to obtain this understanding by testing the policy directly in the real-world. In such cases we have to rely on observational data collected under a different policy than the target. Using this observational data to evaluate the target policy is known as off-policy evaluation (OPE).

Traditionally, most techniques for OPE in contextual bandits focus on evaluating policies based on their **expected** outcomes; see e.g., [12; 30; 26; 23; 24; 6]. However, this can be problematic as methods that are only concerned with the average outcome do not take into account any notions of variance, for example. Therefore, in risk-sensitive settings such as econometrics, where we want to minimize the potential risks, metrics such as CVaR (Conditional Value at Risk) might be more appropriate [11]. Additionally, when only small sample sizes of observational data are available, the average outcomes under finite data can be misleading, as they are prone to outliers and hence, metrics such as medians or quantiles are more robust in these scenarios [1].

Notable exceptions in the OPE literature are [9; 5]. Instead of estimating bounds on the expected outcomes, [9; 5] establish finite-sample bounds for a general class of metrics (e.g., Mean, CVaR, CDF) on the outcome. Their methods can be used to estimate quantiles of the outcomes under the target policy and are therefore robust to outliers. However, the resulting bounds do not depend on the

---

*\*Denotes equal contribution, where ordering was determined through coin flip. Corresponding authors muhammad.taufiq@stats.ox.ac.uk and jeanfrancois@bytedance.com.*

36th Conference on Neural Information Processing Systems (NeurIPS 2022).

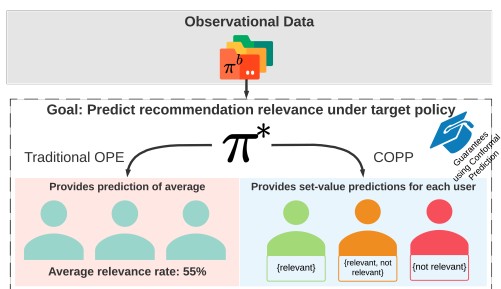 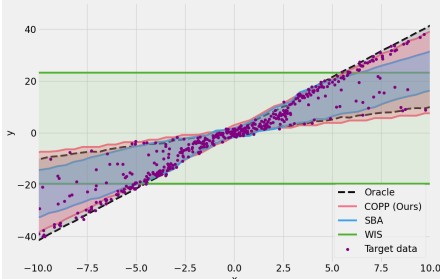

Figure 1: **Left (a):** Conformal Off-Policy Prediction against standard off-policy evaluation methods. **Right (b):** $90\%$ predictive intervals for $Y$ against $X$ for COPP, competing methods and the oracle.

covariates $X$ (not adaptive w.r.t. $X$). This can lead to overly conservative intervals, as we will show in our experiments and can become uninformative when the data are heteroscedastic (see Fig. 1b).

In this paper, we propose Conformal Off-Policy Prediction (COPP), a novel algorithm that uses Conformal Prediction (CP) [29] to construct predictive interval/sets for outcomes in contextual bandits (see Fig. 1a) using an observational dataset. COPP enjoys both finite-sample theoretical guarantees and adaptivity w.r.t. the covariates $X$, and, to the best of our knowledge, is the first such method based on CP that can be applied to stochastic policies and continuous action spaces.

In summary, our contributions are: (i) We propose an application of CP to construct predictive intervals for bandit outcomes that is more general (applies to stochastic policies and continuous actions) than previous work. (ii) We provide theoretical guarantees for COPP, including finite-sample guarantees on marginal coverage and asymptotic guarantees on conditional coverage. (iii) We show empirically that COPP outperforms standard methods in terms of coverage and predictive interval width when assessing new policies.

## 1.1 Problem Setup

Let $\mathcal{X}$ be the covariate space (e.g., user data), $\mathcal{A}$ the action space (e.g., recommended items) and $\mathcal{Y}$ the outcome space (e.g., relevance to the user). We allow both $\mathcal{A}$ and $\mathcal{Y}$ to be either discrete or continuous. In our setting, we are given logged observational data $\mathcal{D}_{obs} = \{x_i, a_i, y_i\}_{i=1}^{n_{obs}}$ where actions are sampled from a behavioural policy $\pi^b$, i.e. $A \mid x \sim \pi^b(\cdot \mid x)$ and $Y \mid x, a \sim P(\cdot \mid x, a)$. We assume that we do not suffer from unmeasured confounding. At test time, we are given a state $x^{test}$ and a new policy $\pi^*$. While $\pi^b$ may be unknown, we assume the target policy $\pi^*$ to be known.

We consider the task of rigorously quantifying the performance of $\pi^*$ without any distributional assumptions on $X$ or $Y$. Many existing approaches estimate $\mathbb{E}_{\pi^*}[Y]$, which is useful for comparing two policies directly as they return a single number. However, the expectation does not convey fine-grained information about how the policy performs for a specific value of $X$, nor does it account for the uncertainty in the outcome $Y$.

Here, we aim to construct intervals/sets on the outcome $Y$ which are (i) adaptive w.r.t. $X$, (ii) capture the variability in the outcome $Y$ and (iii) provide finite-sample guarantees. Current methods lack at least one of these properties (see Sec. 5). One way to achieve these properties is to construct a set-valued function of $x$, $\hat{C}(x)$ which outputs a *subset* of $\mathcal{Y}$. Given any finite dataset $\mathcal{D}_{obs}$, this subset is guaranteed to contain the true value of $Y$ with any pre-specified probability, i.e.

$$1 - \alpha \le \mathbb{P}_{(X,Y) \sim P_{X,Y}^{\pi^*}}(Y \in \hat{C}(X)) \le 1 - \alpha + o_{n_{obs}}(1) \tag{1}$$

where $n_{obs}$ is the size of available observational data and $P_{X,Y}^{\pi^*}$ is the joint distribution of $(X, Y)$ under target policy $\pi^*$. In practice, $\hat{C}(x)$ can be used as a diagnostic tool downstream for a granular assessment of likely outcomes under a target policy. The probability in (1) is taken over the joint distribution of $(X, Y)$, meaning that (1) holds marginally in $X$ (marginal coverage) and not for a given $X = x$ (conditional coverage). In Sec. 4.2, we provide additional regularity conditions under which not only marginal but also conditional coverage holds. Next, we introduce the Conformal Prediction framework, which allows us to construct intervals $\hat{C}(x)$ that satisfy (1) along with properties (i)-(iii).

## 2  Background

Conformal prediction [29; 19] is a methodology that was originally used to compute distribution-free prediction sets for regression and classification tasks. Before introducing COPP, which applies CP to contextual bandits, we first illustrate how CP can be used in standard regression.

### 2.1  Standard Conformal Prediction

Consider the problem of regressing $Y \in \mathcal{Y}$ against $X \in \mathcal{X}$. Let $\hat{f}$ be a model trained on the *training* data $\mathcal{D}_{tr} = \{X_i^0, Y_i^0\}_{i=1}^m \overset{\text{i.i.d.}}{\sim} P_{X,Y}$ and let the *calibration* data $\mathcal{D}_{cal} = \{X_i, Y_i\}_{i=1}^n \overset{\text{i.i.d.}}{\sim} P_{X,Y}$ be independent of $\mathcal{D}_{tr}$. Given a desired coverage rate $1 - \alpha \in (0, 1)$, we construct a band $\hat{C}_n : \mathcal{X} \to \{\text{subsets of } \mathcal{Y}\}$, based on the calibration data such that, for a new i.i.d. test data $(X, Y) \sim P_{X,Y}$,

$$1 - \alpha \leq \mathbb{P}_{(X,Y) \sim P_{X,Y}}(Y \in \hat{C}_n(X)) \leq 1 - \alpha + \frac{1}{n+1}, \tag{2}$$

where the probability is taken over $X, Y$ and $\mathcal{D}_{cal} = \{X_i, Y_i\}_{i=1}^n$ and is conditional upon $\mathcal{D}_{tr}$.

In order to obtain $\hat{C}_n$ satisfying (2), we introduce a non-conformity score function $V_i = s(X_i, Y_i)$, e.g., $(\hat{f}(X_i) - Y_i)^2$. We assume here $\{V_i\}_{i=1}^n$ have no ties almost surely. Intuitively, the non-conformity score $V_i$ uses the outputs of the predictive model $\hat{f}$ on the calibration data, to measure how far off these predictions are from the ground truth response. Higher scores correspond to worse fit between $x$ and $y$ according to $\hat{f}$. We define the empirical distribution of the scores $\{V_i\}_{i=1}^n \cup \{\infty\}$

$$\hat{F}_n := \frac{1}{n+1} \sum_{i=1}^n \delta_{V_i} + \frac{1}{n+1} \delta_\infty \tag{3}$$

with which we can subsequently construct the conformal interval $\hat{C}_n$ that satisfies (2) as follows:

$$\hat{C}_n(x) := \{y : s(x, y) \leq \eta\} \tag{4}$$

where $\eta$ is an empirical quantile of $\{V_i\}_{i=1}^n$, i.e. $\eta = \text{Quantile}_{1-\alpha}(\hat{F}_n)$ is the $1 - \alpha$ quantile.

Intuitively, for roughly $100 \cdot (1 - \alpha)\%$ of the calibration data, the score values will be below $\eta$. Therefore, if the new datapoint $(X, Y)$ and $\mathcal{D}_{cal}$ are i.i.d., the probability $\mathbb{P}(s(X, Y) \leq \eta)$ (which is equal to $\mathbb{P}(Y \in \hat{C}_n(X))$ by (4)) will be roughly $1 - \alpha$. Exchangeability of the data is crucial for the above to hold. In the next section we will explain how [27] relax the exchangeability assumption.

### 2.2  Conformal Prediction under covariate shift

[27] extend the CP framework beyond the setting of exchangeable data, by constructing valid intervals even when the calibration data and test data are not drawn from the same distribution. The authors focus on the *covariate shift* scenario i.e. the distribution of the covariates changes at test time:

$$(X_i, Y_i) \overset{\text{i.i.d}}{\sim} P_{X,Y} = P_X \times P_{Y|X}, \quad i = 1, \ldots, n$$
$$(X, Y) \sim \tilde{P}_{X,Y} = \tilde{P}_X \times P_{Y|X}, \text{ independently}$$

where the ratio $w(x) := \mathrm{d}\tilde{P}_X / \mathrm{d}P_X(x)$ is known. The key realization in [27] is that the requirement of *exchangeability* in CP can be relaxed to a more general property, namely *weighted exchangeability* (see Def. A.1). They propose a weighted version of conformal prediction, which shifts the empirical distribution of non-conformity scores, $\hat{F}_n$, at a point $x$, using weights $w(x)$. This adjusts $\hat{F}_n$ for the covariate shift, before picking the quantile $\eta$:

$$\hat{F}_n^x := \sum_{i=1}^n p_i^w(x) \delta_{V_i} + p_{n+1}^w(x) \delta_\infty \quad \text{where,}$$

$$p_i^w(x) = \frac{w(X_i)}{\sum_{j=1}^n w(X_j) + w(x)}, p_{n+1}^w(x) = \frac{w(x)}{\sum_{j=1}^n w(X_j) + w(x)}.$$

In standard CP (without covariate shift), the weight function satisfies $w(x) = 1$ for all $x$, and we recover (3). Next, we construct the conformal prediction intervals $\hat{C}_n$ as in standard CP using (4) where $\eta$ now depends on $x$ due to $p_i^w(x)$. The resulting intervals, $\hat{C}_n$, satisfy:

$$\mathbb{P}_{(X,Y) \sim \tilde{P}_{X,Y}}(Y \in \hat{C}_n(X)) \geq 1 - \alpha$$

As mentioned previously in Sec. 1.1, the above demonstrates marginal coverage guarantees over test point $X$ and calibration dataset $\mathcal{D}_{cal}$, not conditional on a given $X = x$ or a fixed $\mathcal{D}_{cal}$. We will discuss this nuance later on in Sec. 4.2. In addition, previous work by Vovk [28] shows that conditioned on a single calibration dataset, standard CP can achieve coverage that is 'close' to the required coverage with high probability. However, this has not been extended to the case where the distribution shifts. This is out of the scope of this paper and an interesting future direction.

---

**Algorithm 1:** Conformal Off-Policy Prediction (COPP)

---

**Inputs:** Observational data $\mathcal{D}_{obs} = \{X_i, A_i, Y_i\}_{i=1}^{n_{obs}}$, conf. level $\alpha$, a score function $s(x,y) \in \mathbb{R}$, new data point $x^{test}$, target policy $\pi^*$ ;

**Output:** Predictive interval $\hat{C}_n(x^{test})$;

Split $\mathcal{D}_{obs}$ into training data ($\mathcal{D}_{tr}$) and calibration data ($\mathcal{D}_{cal}$) of sizes $m$ and $n$ respectively;

Use $\mathcal{D}_{tr}$ to estimate weights $\hat{w}(\cdot, \cdot)$ using (7);

Compute $V_i := s(X_i, Y_i)$ for $(X_i, A_i, Y_i) \in \mathcal{D}_{cal}$;

Let $\hat{F}_n^{x,y}$ be the weighted distribution of scores $\hat{F}_n^{x,y} := \sum_{i=1}^n p_i^{\hat{w}}(x,y)\delta_{V_i} + p_{n+1}^{\hat{w}}(x,y)\delta_\infty$

where $p_i^{\hat{w}}(x,y) = \frac{\hat{w}(X_i,Y_i)}{\sum_{j=1}^n \hat{w}(X_j,Y_j)+\hat{w}(x,y)}$ and $p_{n+1}^{\hat{w}}(x,y) = \frac{\hat{w}(x,y)}{\sum_{j=1}^n \hat{w}(X_j,Y_j)+\hat{w}(x,y)}$;

For $x^{test}$ construct: $\hat{C}_n(x^{test}) := \{y : s(x^{test}, y) \leq \text{Quantile}_{1-\alpha}(\hat{F}_n^{x^{test},y})\}$

**Return** $\hat{C}_n(x^{test})$

---

Thus [27] show that the CP algorithm can be extended to the setting of covariate shift with the resulting predictive intervals satisfying the coverage guarantees when the weights are known. The extension of these results to approximate weights was proposed in [14] and is generalized to our setting in Sec. 4.

## 3   Conformal Off-Policy Prediction (COPP)

In the contextual bandits introduced in Sec. 1.1, we assume that the observational data $\mathcal{D}_{obs} = \{x_i, a_i, y_i\}_{i=1}^{n_{obs}}$ is generated from a behavioural policy $\pi^b$. At inference time we are given a new target policy $\pi^*$ and want to provide intervals on the outcomes $Y$ for covariates $X$ that satisfy (1).

The key insight of our approach is to consider the following joint distribution of $(X, Y)$:

$$P^{\pi^b}(x,y) = P(x) \int P(y|x,a)\pi^b(a|x)\mathrm{d}a = P(x)P^{\pi^b}(y|x)$$

$$P^{\pi^*}(x,y) = P(x) \int P(y|x,a)\pi^*(a|x)\mathrm{d}a = P(x)P^{\pi^*}(y|x)$$

Therefore, the change of policies from $\pi^b$ to $\pi^*$ causes a shift in the joint distributions of $(X, Y)$ from $P_{X,Y}^{\pi^b}$ to $P_{X,Y}^{\pi^*}$. More precisely, a shift in the conditional distribution of $Y|X$. As a result, our problem boils down to using CP in the setting where the conditional distribution $P_{Y|X}^{\pi^b}$ changes to $P_{Y|X}^{\pi^*}$ due to the different policies, while the covariate distribution $P_X$ remains the same.

Hence our problem is not concerned about covariate shift as addressed in [27], but instead uses the idea of *weighted exchangeability* to extend CP to the setting of policy shift. To account for this distributional mismatch, our method shifts the empirical distribution of non-conformity scores at a point $(x, y)$ using the weights $w(x,y) = \mathrm{d}P_{X,Y}^{\pi^*}/\mathrm{d}P_{X,Y}^{\pi^b}(x,y) = \mathrm{d}P_{Y|X}^{\pi^*}/\mathrm{d}P_{Y|X}^{\pi^b}(x,y)$:

$$\hat{F}_n^{x,y} := \sum_{i=1}^n p_i^w(x,y)\delta_{V_i} + p_{n+1}^w(x,y)\delta_\infty, \tag{5}$$

$$p_i^w(x,y) = \frac{w(X_i,Y_i)}{\sum_{j=1}^n w(X_j,Y_j)+w(x,y)}, \quad p_{n+1}^w(x,y) = \frac{w(x,y)}{\sum_{j=1}^n w(X_j,Y_j)+w(x,y)}.$$

The intervals are then constructed as below which we call Conformal Off-Policy Prediction (Alg. 1).

$$\hat{C}_n(x^{test}) := \{y : s(x^{test}, y) \leq \eta(x^{test}, y)\} \quad \text{where,} \quad \eta(x,y) := \text{Quantile}_{1-\alpha}(\hat{F}_n^{x,y}). \tag{6}$$

**Remark.** The weights $w(x, y)$ in (5) depend on $x$ and $y$, as opposed to only $x$. In particular, finding the set of $y$'s satisfying (6) becomes more complicated than for the standard covariate shifted CP which only requires a single computation of $\eta(x)$ for a given $x$ as shown in (4). In our case however, we have to create a $k$ sized grid of potential values of $y$ for every $x$ to find $\hat{C}_n(x)$. This operation is embarrassingly parallel and hence does not add much computational overhead compared to the standard CP, especially because CP mainly focuses on scalar predictions.

## 3.1 Estimation of weights $w(x, y)$

So far we have been assuming that we know the weights $w(x, y)$ exactly. However, in most real-world settings, this will not be the case. Therefore, we must resort to estimating $w(x, y)$ using observational data. In order to do so, we first split the observational data into training ($\mathcal{D}_{tr}$) and calibration ($\mathcal{D}_{cal}$) data. Next, using $\mathcal{D}_{tr}$, we estimate $\hat{\pi}^b(a \mid x) \approx \pi^b(a \mid x)$ and $\hat{P}(y \mid x, a) \approx P(y \mid x, a)$ (which is independent of the policy). We then compute a Monte Carlo estimate of weights using the following:

$$\hat{w}(x, y) = \frac{\frac{1}{h} \sum_{k=1}^{h} \hat{P}(y|x, A_k^*)}{\frac{1}{h} \sum_{k=1}^{h} \hat{P}(y|x, A_k)} \approx \frac{\int P(y|x, a)\pi^*(a|x)\mathrm{d}a}{\int P(y|x, a)\pi^b(a|x)\mathrm{d}a}, \tag{7}$$

where $A_k \sim \hat{\pi}^b(\cdot \mid x)$, $A_k^* \sim \pi^*(\cdot \mid x)$ and $h$ is the number of Monte Carlo samples.

**Why not construct intervals using $\hat{P}(y|x, a)$ directly?** We could directly construct predictive intervals $\hat{C}_n(x)$ over outcomes by sampling $Y_j \overset{\text{i.i.d.}}{\sim} \hat{P}^{\pi^*}(y|x) = \int \hat{P}(y|x, a)\pi^*(a|x)\mathrm{d}a$. However, the coverage of these intervals directly depends on the estimation error of $\hat{P}(y|x, a)$. This is not the case in COPP, as the coverage does not depend on $\hat{P}(y|x, a)$ directly but rather on the estimation of $\hat{w}(x, y)$ (see Prop. 4.2). We hypothesize that this indirect dependence of COPP on $\hat{P}(y|x, a)$ makes it less sensitive to the estimation error. In Sec. 6, our empirical results support this hypothesis as COPP provides more accurate coverage than directly using $\hat{P}(y|x, a)$ to construct intervals. Lastly, in Appendix B.5.2 we show how we can avoid estimating $\hat{P}(y|x, a)$ by proposing an alternative method for estimating the weights directly. We leave this for future work.

# 4 Theoretical Guarantees

## 4.1 Marginal Coverage

In this section we provide theoretical guarantees on marginal coverage $\mathbb{P}_{(X,Y) \sim P_{X,Y}^{\pi^*}}(Y \in \hat{C}_n(X))$ for the cases where the weights $w(x, y)$ are known exactly as well as when they are estimated. Using the idea of *weighted exchangeability*, we extend [27, Theorem 2] to our setting.

**Proposition 4.1.** *Let $\{X_i, Y_i\}_{i=1}^n \overset{\text{i.i.d.}}{\sim} P_{X,Y}^{\pi^b}$ be the calibration data. For any score function $s$, and any $\alpha \in (0, 1)$, define the conformal predictive interval at a point $x \in \mathbb{R}^d$ as $\hat{C}_n(x) := \{y \in \mathbb{R} : s(x, y) \le \eta(x, y)\}$ where $\eta(x, y) := \text{Quantile}_{1-\alpha}(\hat{F}_n^{x,y})$, and $\hat{F}_n^{x,y}$ is as defined in (5) with exact weights $w(x, y)$. If $P^{\pi^*}(y|x)$ is absolutely continuous w.r.t. $P^{\pi^b}(y|x)$, then $\hat{C}_n$ satisfies $\mathbb{P}_{(X,Y) \sim P_{X,Y}^{\pi^*}}(Y \in \hat{C}_n(X)) \ge 1 - \alpha$.*

Proposition 4.1 assumes exact weights $w(x, y)$, which is usually not the case. For CP under covariate shift, [14] showed that even when the weights are approximated, i.e., $\hat{w}(x, y) \neq w(x, y)$, we can still provide finite-sample upper and lower bounds on the coverage, albeit slightly modified with an error term $\Delta_w$ (see (8)). We show that this result can be extended to our setting when the weight function $w(x, y)$ is approximated as in Sec. 3.1.

**Proposition 4.2.** *Let $\hat{C}_n$ be the conformal predictive intervals obtained as in Proposition 4.1 with weights $w(x, y)$ replaced by approximate weights $\hat{w}(x, y) = \hat{w}(x, y; \mathcal{D}_{tr})$, where the training data, $\mathcal{D}_{tr}$, is fixed. Assume that $\hat{w}(x, y)$ satisfies $(\mathbb{E}_{(X,Y) \sim P_{X,Y}^{\pi^b}}[\hat{w}(X, Y)^r])^{1/r} \le M_r < \infty$ for some $r \ge 2$. Define $\Delta_w$ as,*

$$\Delta_w := \tfrac{1}{2}\mathbb{E}_{(X,Y) \sim P_{X,Y}^{\pi^b}} \mid \hat{w}(X, Y) - w(X, Y) \mid . \tag{8}$$

*Then,* $\mathbb{P}_{(X,Y) \sim P_{X,Y}^{\pi^*}}(Y \in \hat{C}_n(X)) \ge 1 - \alpha - \Delta_w.$

*If, in addition, non-conformity scores $\{V_i\}_{i=1}^n$ have no ties almost surely, then we also have*

$$\mathbb{P}_{(X,Y)\sim P_{X,Y}^{\pi^*}}(Y \in \hat{C}_n(X)) \leq 1 - \alpha + \Delta_w + cn^{1/r-1},$$

*for some positive constant $c$ depending only on $M_r$ and $r$.*

Proposition 4.2 provides finite-sample guarantees with approximate weights $\hat{w}(\cdot, \cdot)$. Note that if the weights are known exactly then the above proposition can be simplified by setting $\Delta_w = 0$. In the case where the weight function is estimated *consistently*, we recover the exact coverage asymptotically. A natural question to ask is whether the consistency of $\hat{w}(x, y)$ implies the consistency of $\hat{P}(y|x, a)$; in which case one could use $\hat{P}(y|x, a)$ directly to construct the intervals. We prove that this is not the case in general and provide detailed discussion in Appendix B.5.

## 4.2 Conditional Coverage

So far we only considered marginal coverage (1), where the probability is over both $X$ and $Y$. Here, we provide results on conditional coverage $\mathbb{P}_{Y \sim P_{Y|X}^{\pi^*}}(Y \in \hat{C}_n(X) \mid X)$ which is a strictly stronger notion of coverage than marginal coverage [8]. [28; 13] prove that exact conditional coverage cannot be achieved without making additional assumptions. However, we show that, in the case where $Y$ is a continuous random variable and we can estimate the quantiles of $P_{Y|X}^{\pi^*}$ consistently, we get an approximate conditional coverage guarantee using the below proposition.

**Proposition 4.3** (Asymptotic Conditional Coverage). *Let $m, n$ be the number of training and calibration data respectively, $\hat{q}_{\beta,m}(x) = \hat{q}_{\beta,m}(x; \mathcal{D}_{tr})$ be an estimate of the $\beta$-th conditional quantile $q_\beta(x)$ of $P_{Y|X=x}^{\pi^*}$, $\hat{w}_m(x, y) = \hat{w}_m(x, y; \mathcal{D}_{tr})$ be an estimate of $w(x, y)$ and $\hat{C}_{m,n}(x)$ be the conformal interval resulting from algorithm 1 with score function $s(x, y) = \max\{y - \hat{q}_{\alpha_{hi}}(x), \hat{q}_{\alpha_{lo}}(x) - y\}$ where $\alpha_{hi} - \alpha_{lo} = 1 - \alpha$. Assume that the following hold:*

1. *$\lim_{m\to\infty} \mathbb{E}_{(X,Y)\sim P_{X,Y}^{\pi^b}} |\hat{w}_m(X, Y) - w(X, Y)| = 0$.*

2. *there exists $r, b_1, b_2 > 0$ such that $P^{\pi^*}(y \mid x) \in [b_1, b_2]$ uniformly over all $(x, y)$ with $y \in [q_{\alpha_{lo}}(x) - r, q_{\alpha_{lo}}(x) + r] \cup [q_{\alpha_{hi}}(x) - r, q_{\alpha_{hi}}(x) + r]$,*

3. *$\exists k > 0$ s.t. $\lim_{m\to\infty} \mathbb{E}_{X\sim P_X}[H_m^k(X)] = 0$ where $H_m(x) = \max\{|\hat{q}_{\alpha_{lo},m}(x) - q_{\alpha_{lo}}(x)|, |\hat{q}_{\alpha_{hi},m}(x) - q_{\alpha_{hi}}(x)|\}$*

*Then for any $t > 0$, we have that $\lim_{m,n\to\infty} \mathbb{P}(\mathbb{P}_{Y\sim P_{Y|X}^{\pi^*}}(Y \in \hat{C}_{m,n}(X) \mid X) \leq 1 - \alpha - t) = 0$.*

**Remark.** One caveat of Prop. 4.3 is that Assumption 3 is rather strong. In general, consistently estimating the quantiles under the target policy $\pi^*$ is not straightforward given that we only have access to observational data from $\pi^b$. While one can use a weighted pinball loss to estimate quantiles under $\pi^*$, consistent estimation of these quantiles would require a consistent estimate of the weights (see Appendix C). Hence, unlike [14, Theorem 1], our Prop. 4.3 is not a "*doubly robust*" result.

**Towards Group Balanced Coverage.** As pointed out by [2], we may want predictive intervals that have the same coverage across different groups, e.g., across male and female users [17]. Standard CP will not necessarily achieve this, as the coverage guarantee (1) is over the entire population of users. However, we can use COPP on each subgroup separately to obtain group balanced coverage. A more detailed discussion on how to construct such intervals has been included in Appendix B.4.

## 5 Related Work

**Conformal Prediction:** A number of works have explored the use of CP under distribution shift. The works of [27] and [14] are particularly notable as they extend CP to the general setting of *weighted exchangeability*. In particular, [14] use CP for counterfactual inference where the goal is to obtain predictive intervals on the outcomes of treatment and control groups. The authors formulate the counterfactual setting into that of covariate shift in the input space $\mathcal{X}$ and show that under certain assumptions, finite-sample coverage can be guaranteed.

Table 1: Toy experiment results with required coverage 90%. While WIS intervals provide required coverage, the mean interval length is huge compared to COPP (see table 1b).

(a) Mean Coverage as a function of policy shift with 2 standard errors over 10 runs.

| Coverage | $\Delta_\epsilon = 0.0$ | $\Delta_\epsilon = 0.1$ | $\Delta_\epsilon = 0.2$ |
|---|---|---|---|
| COPP (Ours) | **0.90 ± 0.01** | **0.90 ± 0.01** | **0.91 ± 0.01** |
| WIS | 0.89 ± 0.01 | 0.91 ± 0.02 | 0.94 ± 0.02 |
| SBA | 0.90 ± 0.01 | 0.88 ± 0.01 | 0.87 ± 0.01 |
| COPP (GT weights Ours) | **0.90 ± 0.01** | **0.90 ± 0.01** | 0.90 ± 0.01 |
| CP (no policy shift) | 0.90 ± 0.01 | 0.87 ± 0.01 | 0.85 ± 0.01 |
| CP (union) | 0.96 ± 0.01 | 0.96 ± 0.01 | 0.96 ± 0.01 |

(b) Mean Interval Length as a function of policy shift with 2 standard errors over 10 runs.

| Interval Lengths | $\Delta_\epsilon = 0.0$ | $\Delta_\epsilon = 0.1$ | $\Delta_\epsilon = 0.2$ |
|---|---|---|---|
| COPP (Ours) | 9.08 ± 0.10 | 9.48 ± 0.22 | 9.97 ± 0.38 |
| WIS | 24.14 ± 0.30 | 32.96 ± 1.80 | 43.12 ± 3.49 |
| SBA | 8.78 ± 0.12 | 8.94 ± 0.10 | 8.33 ± 0.09 |
| COPP (GT weights Ours) | 8.91 ± 0.09 | 9.25 ± 0.12 | 9.59 ± 0.20 |
| CP (no policy shift) | 9.00 ± 0.10 | 9.00 ± 0.10 | 9.00 ± 0.10 |
| CP (union) | 10.66 ± 0.18 | 11.04 ± 0.2 | 11.4 ± 0.26 |

Fundamentally, our work differs from [14] by framing the problem as a shift in the conditional $P_{Y|X}$ rather than as a shift in the marginal $P_X$. The resulting methodology we obtain from this then differs from [14] in a variety of ways. For example, while [14] assume a deterministic target policy, COPP can also be applied to stochastic target policies, which have been used in a variety of applications, such as recommendation systems or RL applications [25; 22; 7]. Likewise, unlike [14], COPP is applicable to continuous action spaces, e.g., doses of medication administered.

In addition, when the target policy is deterministic, there is an important methodological difference between COPP and [14]. In particular, [14] construct the intervals on outcomes by splitting calibration data w.r.t. actions. In contrast, it can be shown that COPP uses the entire calibration data when constructing intervals on outcomes. This is a consequence of integrating out the actions in the weights $w(x, y)$ (7), and empirically leads to smaller variance in coverage compared to [14]. See B.1 for the experimental results comparing COPP to [14] for deterministic policies.

[15] propose using CP to *construct* robust policies in contextual bandits with discrete actions. Their methodology uses CP to choose actions and does not involve evaluating target policies. Hence, the problem being considered is orthogonal to ours. There has also been concurrent work adapting CP to individual treatment effect (ITE) sensitivity analysis model [10; 32]. Similar to our approach, these works formulate the sensitivity analysis problem as one of CP under the joint distribution shift $P_{X,Y}$. While our methodologies are related, the application of CP explored in these works, i.e. ITE estimation under unobserved confounding, is fundamentally different.

**Uncertainty in contextual bandits:** Recall from the introduction, that most works in this area have focused on quantifying uncertainty in expected outcome (policy value) [6; 12]. Despite providing finite sample-guarantees on the expectation, these methods do not account for the variability in the outcome itself and in general are not adaptive w.r.t. $X$, i.e. they do not satisfy properties (i), (ii) from Sec. 1.1. [9; 5] on the other hand, propose off-policy assessment algorithms for contextual bandits w.r.t. a more general class of risk objectives such as Mean, CVaR etc. Their methodologies can be applied to our problem, to construct predictive intervals for off-policy outcomes. However, unlike COPP, these intervals are not adaptive w.r.t. $X$, i.e. do not satisfy property (i) in Sec. 1.1. Moreover, they do not provide upper bounds on coverage probability, which often leads to overly conservative intervals, as shown in our experiments. Lastly, while distributional perspective has been explored in reinforcement learning [3], no finite sample-guarantees are available to the best of our knowledge.

# 6 Experiments

**Baselines for comparison.** Given our problem setup, there are no established baselines. Instead, we compare our proposed method COPP to the following competing methods, which were constructed to capture the uncertainty in the outcome distribution and take into account the policy shift.

**Weighted Importance Sampling (WIS) CDF estimator.** Given observational dataset $\mathcal{D}_{obs} = \{x_i, a_i, y_i\}_{i=1}^{n_{obs}}$, [9] proposed a non-parametric WIS-based estimator for the empirical CDF of $Y$ under $\pi^*$, $\hat{F}_{WIS}(t) \coloneqq \frac{\sum_{i=1}^{n_{obs}} \hat{\rho}(a_i, x_i) \mathbb{1}(y_i \leq t)}{\sum_{i=1}^{n_{obs}} \hat{\rho}(a_i, x_i)}$ where $\hat{\rho}(a, x) \coloneqq \frac{\pi^*(a|x)}{\hat{\pi}^b(a|x)}$ are the importance weights. We can use $\hat{F}_{WIS}$ to get predictive intervals $[y_{\alpha/2}, y_{1-\alpha/2}]$ where $y_\beta \coloneqq \text{Quantile}_\beta(\hat{F}_{WIS})$. The intervals $[y_{\alpha/2}, y_{1-\alpha/2}]$ do not depend on $x$.

**Sampling Based Approach (SBA).** As mentioned in Sec. 3.1, we can directly use the estimated $\hat{P}(y \mid x, a)$ to construct the predictive intervals as follows. For a given $x^{test}$, we generate $A_i \overset{\text{i.i.d.}}{\sim} \pi^*(\cdot \mid x^{test})$, and $Y_i \sim \hat{P}(\cdot \mid x^{test}, A_i)$ for $i \leq \ell$. We then define the predictive intervals for $x^{test}$ using the $\alpha/2$ and $1 - \alpha/2$ quantiles of $\{Y_i\}_{i \leq \ell}$. While SBA is not a standard baseline, it is a natural comparison to make to answer the question of "why not construct the intervals using $\hat{P}(y|x,a)$ directly"?

## 6.1 Toy Experiment

We start with synthetic experiments and an ablation study, in order to dissect and understand our proposed methodology in more detail. We assume that our policies are stationary and there is overlap between the behaviour and target policy, both of which are standard assumptions [9; 21; 31].

### 6.1.1 Synthetic data experiments setup

In order to understand how COPP works, we construct a simple experimental setup where we can control the amount of "*policy shift*" and know the ground truth. In this experiment, $X \in \mathbb{R}$, $A \in \{1, 2, 3, 4\}$ and $Y \in \mathbb{R}$, where $X$ and $Y \mid x, a$ are normal random variables. Further details and additional experiments on continuous action spaces are given in Appendix D.1.

**Behaviour and Target Policies.** We define a family of policies $\pi_\epsilon(a \mid x)$, where we use the parameter $\epsilon \in (0, 1/3)$ to control the policy shift between target and behaviour policies. Exact form of $\pi_\epsilon(a \mid x)$ is given in D.1. For the behaviour policy $\pi^b$, we use $\epsilon^b = 0.3$ (i.e. $\pi^b(a \mid x) \equiv \pi_{0.3}(a \mid x)$), and for target policies $\pi^*$, we use $\epsilon^* \in \{0.1, 0.2, 0.3\}$. Using the true behaviour policy, $\pi^b$, we generate observational data $\mathcal{D}_{obs} = \{x_i, a_i, y_i\}_{i=1}^{n_{obs}}$ which is then split into training ($\mathcal{D}_{tr}$) and calibration ($\mathcal{D}_{cal}$) datasets, of sizes $m$ and $n$ respectively.

**Estimation of ratios, $\hat{w}(x, y)$.** Using the training dataset $\mathcal{D}_{tr}$, we estimate $P(y|x, a)$ as $\hat{P}(y|x, a) = \mathcal{N}(\mu(x, a), \sigma(x, a))$, where $\mu(x, a), \sigma(x, a)$ are both neural networks (NNs). Similarly, we use NNs to estimate the behaviour policy $\hat{\pi}^b$ from $\mathcal{D}_{tr}$. Next, to estimate $\hat{w}(x, y)$, we use (7) with $h = 500$.

**Score.** For the score function, we use the same formulation as in [18], i.e. $s(x, y) = \max\{\hat{q}_{\alpha_{lo}}(x) - y, y - \hat{q}_{\alpha_{hi}}(x)\}$, where $\hat{q}_\beta(x)$ denotes the $\beta$ quantile estimate of $P^{\pi^b}_{Y|X=x}$ trained using pinball loss.

Lastly, our weights $w(x, y)$ depend on $x$ **and** $y$ and hence we use a grid of 100 equally spaced out $y$'s in our experiments to determine the predictive interval which satisfies $\hat{C}_n(x) := \{y : s(x, y) \leq \text{Quantile}_{1-\alpha}(\hat{F}_n^{x,y})\}$. This is parallelizable and hence does not add much computational overhead.

**Results.** Table 1a shows the coverages of different methods as the policy shift $\Delta_\epsilon = \epsilon^b - \epsilon^*$ increases. The behaviour policy $\pi^b = \pi_{0.3}$ is fixed and we use $n = 5000$ calibration datapoints, across 10 runs. Table 1a shows, how COPP stays very close to the required coverage of 90% across all target policies compared to WIS and SBA. WIS intervals are overly conservative i.e. above the required coverage, while the SBA intervals suffer from under-coverage i.e. below the required coverage. These results supports our hypothesis from Sec. 3.1, which stated that COPP is less sensitive to estimation errors of $\hat{P}(y|x,a)$ compared to directly using $\hat{P}(y|x,a)$ for the intervals, i.e. SBA.

Next, Table 1b shows the mean interval lengths and even though WIS has reasonable coverage for $\Delta_\epsilon = 0.0$ and $0.1$, the average interval length is huge compared to COPP. Fig. 1b shows the predictive intervals for one such experiment with $\pi^* = \pi_{0.1}$ and $\pi^b = \pi_{0.3}$. We can see that SBA intervals are overly optimistic, while WIS intervals are too wide and are not adaptive w.r.t. $X$. COPP produces intervals which are much closer to the oracle intervals.

### 6.1.2 Ablation Study.

To isolate the effect of weight estimation error and policy shift, we conduct an ablation study, comparing COPP with estimated weights to COPP with Ground Truth (GT) weights and standard CP (assuming no policy shift). Table 1a shows that at $\Delta_\epsilon = 0$, i.e. no policy shift, standard CP achieves the required coverage as expected. However the coverage of standard CP intervals decreases as the policy shift $\Delta_\epsilon$ increases. COPP, on the other hand, attains the required coverage of 90%, by adapting the predictive intervals with increasing policy shift. Table 1b shows that the average interval length of COPP increases with increasing policy shift $\Delta_\epsilon$. Furthermore, Table 1a illustrates that while COPP achieves the required coverage for different target policies, on average it is slightly more

Table 2: Mean Coverage as a function of policy shift $\Delta_\epsilon$ and 2 standard errors over 10 runs. COPP attains the required coverage of 90%, whereas the competing methods, WIS and SBA, are over-conservative i.e. coverage above 90%. In addition, when we do not account for the policy shift, standard CP becomes progressively worse with increasing policy shift.

| | $\Delta_\epsilon = 0.0$ | $\Delta_\epsilon = 0.1$ | $\Delta_\epsilon = 0.2$ | $\Delta_\epsilon = 0.3$ | $\Delta_\epsilon = 0.4$ |
|---|---|---|---|---|---|
| COPP (Ours) | **0.90 ± 0.00** | **0.90 ± 0.02** | **0.90 ± 0.01** | **0.89 ± 0.01** | **0.91 ± 0.01** |
| WIS | 1.00 ± 0.00 | 1.00 ± 0.00 | 0.92 ± 0.00 | 0.94 ± 0.00 | 0.91 ± 0.00 |
| SBA | 0.99 ± 0.00 | 0.99 ± 0.00 | 0.98 ± 0.00 | 0.97 ± 0.00 | 0.96 ± 0.00 |
| CP (no policy shift) | **0.91 ± 0.02** | **0.92 ± 0.02** | 0.93 ± 0.01 | 0.94 ± 0.01 | 0.96 ± 0.01 |

conservative than using COPP with GT weights. This can be explained by the estimation error in $\hat{w}(x, y)$. Additionally, to investigate the effect of integrating out the actions in (7), we also perform CP for each action $a$ separately (as in [14]) and then take the union of the intervals across these actions. In the union method, the probability of an action being chosen is not taken into account, (i.e., intervals are independent of $\pi^*$) and hence the coverage is overly conservative as expected.

Lastly, we investigate how increasing the number of calibration data $n$ affects the coverage for all the methodologies. We observe that coverage of COPP is closer to the required coverage of 90% compared to the competing methodologies. Additionally, the coverage of COPP converges to the required coverage as $n$ increases; see Appendix D.1.1 for detailed experimental results.

## 6.2 Experiments on Microsoft Ranking Dataset

We now apply COPP onto a real dataset i.e. the Microsoft Ranking dataset 30k [16; 25; 4]. Due to space constraints, we have added additional extensive experiments on UCI datasets in Appendix D.3.

**Dataset.** The dataset contains relevance scores for websites recommended to different users, and comprises of $30,000$ user-website pairs. For each user-website pair, the data contains a 136-dimensional feature vector, which consists of user's attributes corresponding to the website, such as length of stay or number of clicks on the website. Furthermore, for each user-website pair, the dataset also contains a relevance score, i.e. how relevant the website was to the user.

First, given a user, we sample (with replacement) 5 websites from the data corresponding to that user. Next, we reformulate this into a contextual bandit where $a_i \in \{1, 2, 3, 4, 5\}$ corresponds to the action of recommending the $a_i$'th website to the user $i$. $x_i$ is obtained by combining the 5 user-website feature vectors corresponding to the user $i$ i.e. $x_i \in \mathbb{R}^{5 \times 136}$. $y_i \in \{0, 1, 2, 3, 4\}$ corresponds to the relevance score for the $a_i$'th website, i.e. the recommended website. The goal is to construct prediction sets that are guaranteed to contain the true relevance score with a probability of 90%.

**Behaviour and Target Policies.** We first train a NN classifier model, $\hat{f}_\theta$, mapping each 136-dimensional user-website feature vector to the softmax scores for each relevance score class. We use this trained model $\hat{f}_\theta$ to define a family of policies which pick the most relevant website as predicted by $\hat{f}_\theta$ with probability $\epsilon$ and the rest uniformly with probability $(1 - \epsilon)/4$ (see Appendix D.2 for more details). Like the previous experiment, we use $\epsilon$ to control the shift between behaviour and target policies. For $\pi^b$, we use $\epsilon^b = 0.5$ and for $\pi^*$, $\epsilon^* \in \{0.1, 0.2, 0.3, 0.4, 0.5\}$.

**Estimation of ratios** $\hat{w}(X, Y)$. To estimate the $\hat{P}(y \mid x, a)$ we use the trained model $\hat{f}_\theta$ as detailed in Appendix D.2. To estimate the behaviour policy $\hat{\pi}^b$, we train a neural network classifier model $\mathcal{X} \to \mathcal{A}$, and we use (7) to estimate the weights $\hat{w}(x, y)$.

**Score.** The space of outcomes $\mathcal{Y}$ in this experiment is discrete. We define $\hat{P}^{\pi^b}(y \mid x) = \sum_{i=1}^{5} \hat{\pi}^b(A = i | x) \hat{P}(y | x, A = i)$. Using similar formulation as in [2], we define the score:

$$s(x, y) = \sum_{y'=0}^{4} \hat{P}^{\pi^b}(y' \mid x) \mathbb{1}(\hat{P}^{\pi^b}(y' \mid x) \geq \hat{P}^{\pi^b}(y \mid x)).$$

Since $\mathcal{Y}$ is discrete, we no longer need to construct a grid of $y$ values on which to compute $\text{Quantile}_{1-\alpha}(\hat{F}_n^{x,y})$. Instead, we will simply compute this quantity on each $y \in \mathcal{Y}$, when constructing the predictive sets $\hat{C}_n(x^{test})$.

**Results.** Table 2 shows the coverages of different methodologies across varying target policies $\pi_{\epsilon^*}$. The behaviour policy $\pi^b = \pi_{0.5}$ is fixed and we use $n = 5000$ calibration datapoints, across

10 runs. Table 2 also shows that the coverage of WIS and SBA sets is dependent upon the policy shift, with both being overly conservative across the different target policies as compared to COPP. Recall that the WIS sets do not depend on $x^{test}$ and as a result we get the same set for each test data point. This becomes even more problematic when $Y$ is discrete – if, for each label $y$, $\mathbb{P}_{(X,Y) \sim P_{X,Y}^{\pi^*}}(Y = y) > 10\%$, then WIS sets (with the required coverage of 90%) are likely to contain every label $y \in \mathcal{Y}$. In comparison, COPP is able to stay much closer to the required coverage of 90% across all target policies. We have also added standard CP without policy shift as a sanity check, and observed that the sets get increasingly conservative as the policy shift increases.

Finally, we also plotted how the coverage changes as the number of calibration data $n$ increases. We observe again that the coverage of COPP is closer to the required coverage of 90% compared to the competing methodologies. Due to space constraints, we have added the plots in Appendix D.2.

**Class-balanced conformal prediction.** Using the methodology described in Sec. 4.2, we construct predictive sets, $\hat{C}_n^{\mathcal{Y}}(x)$, which offer label conditioned coverage guarantees (see B.4), i.e. for all $y \in \mathcal{Y}$,

$$\mathbb{P}_{(X,Y) \sim P_{X,Y}^{\pi^*}}(Y \in \hat{C}_n^{\mathcal{Y}}(X) \mid Y = y) \geq 1 - \alpha.$$

We empirically demonstrate that $\hat{C}_n^{\mathcal{Y}}$ provides label conditional coverage, while $\hat{C}_n$ obtained using alg. 1 may not. Due to space constraints, details on construction of $\hat{C}_n^{\mathcal{Y}}$ as well as experimental results have been included in Appendix D.2.1.

## 7    Conclusion and Limitations

In this paper, we propose COPP, an algorithm for constructing predictive intervals on off-policy outcomes, which are adaptive w.r.t. covariates $X$. We theoretically prove that COPP can guarantee finite-sample coverage by adapting the framework of conformal prediction to our setup. Our experiments show that conventional methods cannot guarantee any user pre-specified coverage, whereas COPP can. For future work, it would be interesting to apply COPP to policy training. This could be a step towards robust policy learning by optimising the worst case outcome [20].

We conclude by mentioning several limitations of COPP. Firstly, we do not guarantee conditional coverage in general. We outline conditions under which conditional coverage holds asymptotically (Prop. 4.3), however, this relies on somewhat strong assumptions. Secondly, our current method estimates the weights $w(x,y)$ through $P(y \mid x,a)$, which can be challenging. We address this limitation in Appendix B.5.2, where we propose an alternative method to estimate the weights directly, without having to model $P(y \mid x,a)$. Lastly, reliable estimation of our weights $\hat{w}(x,y)$ requires sufficient overlap between behaviour and target policies. The results from COPP may suffer in cases where this assumption is violated, which we illustrate empirically in Appendix D.1.2. We believe these limitations suggest interesting research questions that we leave to future work.

## Acknowledgements

We would like to thanks Andrew Jesson, Sahra Ghalebikesabi, Robert Hu, Siu Lun Chau and Tim Rudner for useful feedback. JFT is supported by the EPSRC and MRC through the OxWaSP CDT programme (EP/L016710/1). MFT acknowledges his PhD funding from Google DeepMind. RC and AD are supported by the Engineering and Physical Sciences Research Council (EPSRC) through the Bayes4Health programme [Grant number EP/R018561/1].

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
