## A  Proofs

### A.1  Proof of Proposition 4.1

This proof is a direct adaptation of [27, Lemma 3], and has only been included for the sake of completeness.

In this proof, we use the notion of *weighted exchangeability* as defined in Section 3.2 of [27].

**Definition A.1** (Weighted exchangeability)**.** Random variables $V_1, \ldots, V_n$ are said to be *weighted exchangeable* with weight functions $w_1, \ldots, w_n$, if the density $f$ of their joint distribution can be factorized as

$$f(v_1, \ldots, v_n) = \prod_{i=1}^{n} w_i(v_i) g(v_1, \ldots, v_n) \tag{9}$$

where $g$ is any function that does not depend on the ordering of its inputs, i.e. $g(v_{\sigma(1)}, \ldots, v_{\sigma(n)}) = g(v_1, \ldots, v_n)$ for any permutation $\sigma$ of $1, \ldots, n$.

**Lemma A.2.** *Let $Z_i = (X_i, Y_i) \in \mathbb{R}^d \times \mathbb{R}$, $i = 1, \ldots, n+1$, be such that $\{(X_i, Y_i)\}_{i=1}^{n} \overset{\text{i.i.d.}}{\sim} P_{X,Y}^{\pi^b}$ and $(X_{n+1}, Y_{n+1}) \sim P_{X,Y}^{\pi^*}$. Then $Z_1, \ldots, Z_{n+1}$ are weighted exchangeable with weights $w_i \equiv 1$, $i \leq n$ and $w_{n+1}(X, Y) = \mathrm{d}P_{X,Y}^{\pi^*}/\mathrm{d}P_{X,Y}^{\pi^b}(X, Y)$.*

*Proof.* The proof below is merely a verification that our proposed weights still retain the coverage guarantees and is mainly taken from [27]. Hence, we follow the same strategy as in [27], with the exception that we have the weights as in Lemma A.2, hence inducing a lot of simplifications. As in [27], we assume for simplicity that $V_1, \ldots, V_{n+1}$ are distinct almost surely, however the result holds in general case as well. We define $f$ as the joint distribution of the random variables $\{X_i, Y_i\}_{i=1}^{n+1}$. We also denote $E_z$ as the event of $\{Z_1, \ldots, Z_{n+1}\} = \{z_1, \ldots, z_{n+1}\}$ and let $v_i = s(z_i) = s(x_i, y_i)$, then for each $i$:

$$\mathbb{P}\{V_{n+1} = v_i | E_z\} = \mathbb{P}\{Z_{n+1} = z_i | E_z\} = \frac{\sum_{\sigma:\sigma(n+1)=i} f(z_{\sigma(1)}, \ldots, z_{\sigma(n+1)})}{\sum_{\sigma} f(z_{\sigma(1)}, \ldots, z_{\sigma(n+1)})} \tag{10}$$

Now using the fact that $Z_1, \ldots, Z_{n+1}$ are weighted exchangeable:

$$\frac{\sum_{\sigma:\sigma(n+1)=i} f(z_{\sigma(1)}, \ldots, z_{\sigma(n+1)})}{\sum_{\sigma} f(z_{\sigma(1)}, \ldots, z_{\sigma(n+1)})} = \frac{\sum_{\sigma:\sigma(n+1)=i} \prod_{j=1}^{n+1} w_j(z_{\sigma(j)}) g(z_{\sigma(1)}, \ldots, z_{\sigma(n+1)})}{\sum_{\sigma} \prod_{j=1}^{n+1} w_j(z_{\sigma(j)}) g(z_{\sigma(1)}, \ldots, z_{\sigma(n+1)})} \tag{11}$$

$$= \frac{w_{n+1}(z_i) g(z_1, \ldots, z_{n+1})}{\sum_{j=1}^{n+1} w_{n+1}(z_j) g(z_1, \ldots, z_{n+1})}$$

$$= p_i^w(z_{n+1})$$

where we recall that

$$p_i^w(x, y) := \frac{w(X_i, Y_i)}{\sum_{j=1}^{n} w(X_j, Y_j) + w(x, y)}.$$

We get simplifications in (11) due to the weights defined in Lemma A.2, i.e. $w_i \equiv 1$ for $i \leq n$ and $w_{n+1}(x, y) = w(x, y) = \mathrm{d}P_{X,Y}^{\pi^*}/\mathrm{d}P_{X,Y}^{\pi^b}(x, y)$. Next, just as in [27] we can view:

$$V_{n+1} = v_i | E_z \sim \sum_{i=1}^{n+1} p_i^w(z_{n+1}) \delta_{v_i} \tag{12}$$

which implies that:

$$\mathbb{P}\{V_{n+1} \leq \text{Quantile}_\beta(\sum_{i=1}^{n+1} p_i^w(z_{n+1}) \delta_{v_i}) | E_z\} \geq \beta.$$

This is equivalent to

$$\mathbb{P}\{V_{n+1} \leq \text{Quantile}_\beta(\sum_{i=1}^{n+1} p_i^w(Z_{n+1})\delta_{v_i})|E_z\} \geq \beta$$

and, after marginalizing, one has

$$\mathbb{P}\{V_{n+1} \leq \text{Quantile}_\beta(\sum_{i=1}^{n+1} p_i^w(Z_{n+1})\delta_{v_i})\} \geq \beta$$

This is equivalent to the claim in Proposition 4.1. $\qquad\square$

## A.2   Proof of Proposition 4.2

The following proof is an adaptation of [14, Proposition 1] to our setting.

Before detailing the main proof, we introduce a preliminary result which will be used in the proof of Proposition 4.2.

**Lemma A.3.** *Let $\hat{w}(x,y)$ be an estimate of the weights $w(x,y) = \mathrm{d}P_{X,Y}^{\pi^*}/\mathrm{d}P_{X,Y}^{\pi^b}(x,y)$, and $(\mathbb{E}_{(X,Y)\sim P_{X,Y}^{\pi^b}}[\hat{w}(X,Y)^r])^{1/r} \leq M_r < \infty$ for some $r \geq 2$. Let $(X_i, Y_i) \overset{\text{i.i.d.}}{\sim} P_{X,Y}^{\pi^b}$ and $\mathcal{A}$ denote the event that*

$$\sum_{i=1}^{n} \hat{w}(X_i, Y_i) \leq n/2.$$

*Then,*

$$\mathbb{P}(\mathcal{A}) \leq \frac{c_1 M_r^2}{n}$$

*where $c_1$ is an absolute constant, and the probability is taken over $\{X_i, Y_i\}_{i=1}^{n} \overset{\text{i.i.d.}}{\sim} P_{X,Y}^{\pi^b}$.*

### Proof of Lemma A.3

The condition $\mathbb{E}_{(X,Y)\sim P_{X,Y}^{\pi^b}}[\hat{w}(X,Y)^r] < \infty \implies \mathbb{P}_{(X,Y)\sim P_{X,Y}^{\pi^b}}(\hat{w}(X,Y) < \infty) = 1$ and $\mathbb{E}_{(X,Y)\sim P_{X,Y}^{\pi^b}}[\hat{w}(X,Y)] < \infty$. WLOG assume $\mathbb{E}_{(X,Y)\sim P_{X,Y}^{\pi^b}}[\hat{w}(X,Y)] = 1$. Recall that $p_i^{\hat{w}}(x,y) := \frac{\hat{w}(X_i,Y_i)}{\sum_{i=1}^{n}\hat{w}(X_i,Y_i)+\hat{w}(x,y)}$, and therefore, $p_i^{\hat{w}}(x,y)$ are invariant to weight scaling. Since $\mathbb{E}_{(X_i,Y_i)\sim P_{X,Y}^{\pi^b}}[\hat{w}(X_i,Y_i)]^2 \leq M_r^2$ and $\mathbb{E}_{(X_i,Y_i)\sim P_{X,Y}^{\pi^b}}(\hat{w}(X_i,Y_i)) = 1$, using Chebyshev's inequality

$$
\begin{aligned}
\mathbb{P}\left(\sum_{i=1}^{n} \hat{w}(X_i, Y_i) \leq n/2\right) &= \mathbb{P}\left(\sum_{i=1}^{n}(\hat{w}(X_i, Y_i) - 1) \leq -n/2\right) \\
&\leq \mathbb{P}\left(|\sum_{i=1}^{n}(\hat{w}(X_i, Y_i) - 1)| \geq n/2\right) \\
&\leq \frac{4}{n^2}\mathbb{E}\left[\left(\sum_{i=1}^{n}\hat{w}(X_i, Y_i) - \mathbb{E}[\hat{w}(X_i, Y_i)]\right)^2\right] \\
&= \frac{4}{n^2}\left\{n\mathbb{E}|\hat{w}(X_1, Y_1) - \mathbb{E}[\hat{w}(X_1, Y_1)]|^2\right\} \qquad (13) \\
&\leq \frac{16}{n^2}n\mathbb{E}|\hat{w}(X_1, Y_1)|^2 \qquad (14) \\
&\leq \frac{c_1 M_r^2}{n}
\end{aligned}
$$

where to get from (13) to (14) we use:

$$
\begin{aligned}
\mathbb{E}|\hat{w}(X_1, Y_1) - \mathbb{E}[\hat{w}(X_1, Y_1)]|^2 &\leq 2\mathbb{E}\left[\hat{w}(X_1, Y_1)^2 + \mathbb{E}[\hat{w}(X_1, Y_1)]^2\right] \\
&\leq 4\mathbb{E}[\hat{w}(X_1, Y_1)^2].
\end{aligned}
$$

$\square$

We can now prove Proposition 4.2.

*Proof.* The condition $\mathbb{E}_{(X,Y)\sim P_{X,Y}^{\pi^b}}[\hat{w}(X,Y)^r] < \infty \implies \mathbb{P}_{(X,Y)\sim P_{X,Y}^{\pi^b}}(\hat{w}(X,Y) < \infty) = 1$ and $\mathbb{E}_{(X,Y)\sim P_{X,Y}^{\pi^b}}[\hat{w}(X,Y)] < \infty$. WLOG assume $\mathbb{E}_{(X,Y)\sim P_{X,Y}^{\pi^b}}[\hat{w}(X,Y)] = 1$. Let $\tilde{P}_{X,Y}^{\pi^*}$ be a probability measure with

$$\mathrm{d}\tilde{P}_{X,Y}^{\pi^*}(x,y) := \hat{w}(x,y)\mathrm{d}P_{X,Y}^{\pi^b}(x,y)$$

and $(\tilde{X}, \tilde{Y}) \sim \tilde{P}_{X,Y}^{\pi^*}$ that is independent of the data. By Hölder's inequality,

$$\mathbb{E}_{(\tilde{X},\tilde{Y})\sim \tilde{P}_{X,Y}^{\pi^*}}[\hat{w}(\tilde{X},\tilde{Y})] = \int_{\tilde{x},\tilde{y}} \frac{\mathrm{d}\tilde{P}^{\pi^*}(\tilde{x},\tilde{y})}{\mathrm{d}P^{\pi^b}(\tilde{x},\tilde{y})}\mathrm{d}\tilde{P}^{\pi^*}(\tilde{x},\tilde{y})$$
$$= \mathbb{E}_{(X,Y)\sim P_{X,Y}^{\pi^b}}[\hat{w}(X,Y)^2] \le M_r^2 < \infty$$

Note using Proposition 4.1 with $(\tilde{X}, \tilde{Y})$ denoting $(X_{n+1}, Y_{n+1})$ for simplicity

$$\mathbb{P}(\tilde{Y} \in \hat{C}(\tilde{X}, \tilde{Y}))$$
$$= \mathbb{E}_{(\tilde{X},\tilde{Y})\sim \tilde{P}_{X,Y}^{\pi^*}} \left[ \mathbb{P}\left( s(\tilde{X}, \tilde{Y}) \le \text{Quantile}_{1-\alpha}\left( \sum_{i=1}^n p_i^{\hat{w}}(\tilde{X}, \tilde{Y})\delta_{V_i} + p_{n+1}^{\hat{w}}(\tilde{X}, \tilde{Y})\delta_\infty \right) \mid \mathcal{E}(\tilde{V}) \right) \right]$$
(15)

where $\mathcal{E}(\tilde{V})$ denotes the unordered set of $V_1, \ldots, V_{n+1}$. Marginalising over $\{(X_i, Y_i)\}_{i=1}^n$, we obtain

$$(15) \le \mathbb{E}\left( 1 - \alpha + \max_{i\in[n+1]} p_i^{\hat{w}}(\tilde{X}, \tilde{Y}) \right)$$
(16)

where the expectation is over $\{(X_i, Y_i)\}_{i=1}^n \overset{\text{i.i.d.}}{\sim} P_{X,Y}^{\pi^b}$ and $(\tilde{X}, \tilde{Y}) \sim \tilde{P}_{X,Y}^{\pi^*}$. Let $\mathcal{A}$ denote the event that

$$\sum_{i=1}^n \hat{w}(X_i, Y_i) \le n/2.$$

using Lemma A.3 and $\mathbb{E}[\hat{w}(\tilde{X}, \tilde{Y})] \le M_r^2$, we get that

$$\mathbb{E}\left[ \max_{i\in[n+1]} p_i^{\hat{w}}(\tilde{X}, \tilde{Y}) \right] = \mathbb{E}\left[ \frac{\max\{\hat{w}(\tilde{X}, \tilde{Y}), \max_i \hat{w}(X_i, Y_i)\}}{\hat{w}(\tilde{X}, \tilde{Y}) + \sum_{i=1}^n \hat{w}(X_i, Y_i)} \right]$$
$$\le \mathbb{E}\left[ \frac{\max\{\hat{w}(\tilde{X}, \tilde{Y}), \max_i \hat{w}(X_i, Y_i)\}}{\hat{w}(\tilde{X}, \tilde{Y}) + \sum_{i=1}^n \hat{w}(X_i, Y_i)} \mathbb{1}_{\mathcal{A}^C} \right] + \mathbb{P}(\mathcal{A})$$
$$\le \mathbb{E}\left[ \frac{2\max\{\hat{w}(\tilde{X}, \tilde{Y}), \max_i \hat{w}(X_i, Y_i)\}}{n} \mathbb{1}_{\mathcal{A}^C} \right] + \frac{c_1 M_r^2}{n}$$
$$\le \frac{2}{n}\left( \mathbb{E}[\hat{w}(\tilde{X}, \tilde{Y})] + \mathbb{E}\max_i \hat{w}(X_i, Y_i) \right) + \frac{c_1 M_r^2}{n}$$
$$\le \frac{2}{n}\left( \mathbb{E}[\hat{w}(\tilde{X}, \tilde{Y})] + \left( \sum_{i=1}^n \mathbb{E}[\hat{w}(X_i, Y_i)^r] \right)^{1/r} \right) + \frac{c_1 M_r^2}{n}$$
$$\le \frac{2}{n}\left( M_r^2 + n^{1/r} M_r \right) + \frac{c_1 M_r^2}{n}.$$

This implies that

$$\mathbb{P}_{(X,Y)\sim \tilde{P}_{X,Y}^{\pi^*}}(Y \in \hat{C}(X)) \le 1 - \alpha + cn^{1/r-1}$$

for some constant $c$ that only depends on $M_r$ and $r$. Note that

$$|\mathbb{P}_{(X,Y)\sim \tilde{P}_{X,Y}^{\pi^*}}(Y \in \hat{C}(X)) - \mathbb{P}_{(X,Y)\sim P_{X,Y}^{\pi^*}}(Y \in \hat{C}(X))| \le d_{\text{TV}}(\tilde{P}^{\pi^*}, P^{\pi^*})$$
(17)

where $d_{\text{TV}}$ is the total variation norm which satisfies

$$
\begin{aligned}
d_{\text{TV}}(\tilde{P}^{\pi^*}, P^{\pi^*}) =& \frac{1}{2}\int |\hat{w}(x,y)\mathrm{d}P^{\pi^b}(x,y) - \mathrm{d}P^{\pi^*}(x,y)| \\
=& \frac{1}{2}\int |\hat{w}(x,y)\mathrm{d}P^{\pi^b}(x,y) - w(x,y)\mathrm{d}P^{\pi^b}(x,y)| \\
=& \frac{1}{2}\mathbb{E}_{(X,Y)\sim P^{\pi^b}_{X,Y}}[|\hat{w}(X,Y) - w(X,Y)|] = \Delta_w.
\end{aligned}
\tag{18}
$$

Putting together (17) and (18), we get

$$
\mathbb{P}_{(X,Y)\sim P^{\pi^*}_{X,Y}}(Y \in \hat{C}(X)) \leq 1 - \alpha + \Delta_w + cn^{1/r-1}.
\tag{19}
$$

For the lower bound, using Proposition 4.1 we get that

$$
\mathbb{P}_{(\tilde{X},\tilde{Y})\sim \tilde{P}^{\pi^*}_{X,Y}}(\tilde{Y} \in \hat{C}(\tilde{X},\tilde{Y})) = \mathbb{P}\left(s(\tilde{X},\tilde{Y}) \leq \text{Quantile}_{1-\alpha}\left(\sum_{i=1}^{n} p_i^{\hat{w}}(\tilde{X},\tilde{Y})\delta_{V_i} + p_{n+1}^{\hat{w}}(\tilde{X},\tilde{Y})\delta_{\infty}\right)\right)
$$
$$
\geq 1 - \alpha.
\tag{20}
$$

Using (17) we thus obtain

$$
\begin{aligned}
\mathbb{P}_{(X,Y)\sim P^{\pi^*}_{X,Y}}(Y \in \hat{C}(X)) \geq& \mathbb{P}_{(X,Y)\sim \tilde{P}^{\pi^*}_{X,Y}}(Y \in \hat{C}(X)) - d_{TV}(\tilde{P}^{\pi^*}, P^{\pi^*}) \\
\geq& 1 - \alpha - \Delta_w.
\end{aligned}
\tag{21}
$$

$\square$

### A.3 Proof of Proposition 4.3

For notational convenience, we suppress the subscripts $m$ and $n$ in $\hat{q}, \hat{w}, \hat{C}$. Moreover, we use $\hat{w}_i$ to denote $\hat{w}(X_i, Y_i)$ and $\eta(x,y)$ to denote $\text{Quantile}_{1-\alpha}(\sum_{i=1}^{n} \hat{p}_i(x,y)\delta_{V_i} + \hat{p}_{n+1}(x,y)\delta_{\infty})$.

*Proof.* We use $(\tilde{X}, \tilde{Y}) \sim P^{\pi^*}_{X,Y}$ in place of $(X_{n+1}, Y_{n+1})$ and let $\epsilon < r/2$. By the definition of $\hat{C}(\tilde{X})$, we directly have

$$
\begin{aligned}
\mathbb{P}(\tilde{Y} \in \hat{C}(\tilde{X}) \mid \tilde{X}) =& \mathbb{P}(s(\tilde{X},\tilde{Y}) \leq \eta(\tilde{X},\tilde{Y}) \mid \tilde{X}) \\
\geq& \mathbb{P}(s^*(\tilde{X},\tilde{Y}) \leq \eta(\tilde{X},\tilde{Y}) - H(\tilde{X}) \mid \tilde{X})
\end{aligned}
\tag{22}
$$

where $s^*(\tilde{X}, \tilde{Y}) := \max\{\tilde{Y} - q_{\alpha_{hi}}(\tilde{X}), q_{\alpha_{lo}}(\tilde{X}) - \tilde{Y}\}$ and the probability is taken over $\{(X_i, Y_i)\}_{i=1}^{n} \overset{\text{i.i.d.}}{\sim} P^{\pi^b}_{X,Y}$ and $\tilde{Y} \sim P^{\pi^*}_{Y|X=\tilde{X}}$. We then get

$$
\begin{aligned}
(22) \geq& \mathbb{P}(s^*(\tilde{X},\tilde{Y}) \leq -\epsilon - H(\tilde{X}) \mid \tilde{X}) - \mathbb{P}(\eta(\tilde{X},\tilde{Y}) < -\epsilon \mid \tilde{X}) \\
\geq& \mathbb{P}(s^*(\tilde{X},\tilde{Y}) \leq -\epsilon - H(\tilde{X}) \mid \tilde{X})\left(\mathbb{1}(H(\tilde{X}) \leq \epsilon) + \mathbb{1}(H(\tilde{X}) > \epsilon)\right) - \mathbb{P}(\eta(\tilde{X},\tilde{Y}) < -\epsilon \mid \tilde{X})
\end{aligned}
\tag{23}
$$

$$
\begin{aligned}
\geq& \left(\mathbb{P}(s^*(\tilde{X},\tilde{Y}) \leq 0 \mid \tilde{X}) - b_2\{\epsilon + H(\tilde{X})\}\right)\mathbb{1}(H(\tilde{X}) \leq \epsilon) \\
&+ \mathbb{P}(s^*(\tilde{X},\tilde{Y}) \leq -\epsilon - H(\tilde{X}) \mid \tilde{X})\mathbb{1}(H(\tilde{X}) > \epsilon) - \mathbb{P}(\eta(\tilde{X},\tilde{Y}) < -\epsilon \mid \tilde{X})
\end{aligned}
\tag{24}
$$

$$
\begin{aligned}
\geq& \mathbb{P}(s^*(\tilde{X},\tilde{Y}) \leq 0 \mid \tilde{X})\mathbb{1}(H(\tilde{X}) \leq \epsilon) - b_2\{\epsilon + H(\tilde{X})\mathbb{1}(H(\tilde{X}) \leq \epsilon)\} \\
&+ \left(\mathbb{P}(s^*(\tilde{X},\tilde{Y}) \leq 0 \mid \tilde{X}) - \mathbb{P}(s^*(\tilde{X},\tilde{Y}) \in (-\epsilon - H(\tilde{X}), 0))\right)\mathbb{1}(H(\tilde{X}) > \epsilon) \\
&- \mathbb{P}(\eta(\tilde{X},\tilde{Y}) < -\epsilon \mid \tilde{X}) \\
\geq& \mathbb{P}(s^*(\tilde{X},\tilde{Y}) \leq 0 \mid \tilde{X}) - b_2\{\epsilon + H(\tilde{X})\mathbb{1}(H(\tilde{X}) \leq \epsilon)\} - \mathbb{1}(H(\tilde{X}) > \epsilon) \\
&- \mathbb{P}(\eta(\tilde{X},\tilde{Y}) < -\epsilon \mid \tilde{X})
\end{aligned}
\tag{25}
$$

where, to get from (23) to (24), we use the condition $2\epsilon < r$ and Assumption 2

$$\text{(25)} \geq \mathbb{P}(s^*(\tilde{X}, \tilde{Y}) \leq 0 \mid \tilde{X}) - b_2\{\epsilon + H(\tilde{X})\} - \mathbb{1}(H(\tilde{X}) > \epsilon) - \mathbb{P}(\eta(\tilde{X}, \tilde{Y}) < -\epsilon \mid \tilde{X}) \quad (26)$$

$$= 1 - \alpha - b_2\{\epsilon + H(\tilde{X})\} - \mathbb{1}(H(\tilde{X}) > \epsilon) - \mathbb{P}(\eta(\tilde{X}, \tilde{Y}) < -\epsilon \mid \tilde{X}). \quad (27)$$

Next, we derive an upper bound on $\mathbb{P}(\eta(\tilde{X}, \tilde{Y}) < -\epsilon \mid \tilde{X})$. Let $G$ denote the CDF of the random distribution $\sum_{i=1}^n \hat{p}_i(x,y)\delta_{V_i} + \hat{p}_{n+1}(x,y)\delta_\infty$. Then, $\eta(\tilde{X}, \tilde{Y}) < -\epsilon$ implies $G(-\epsilon) \geq 1 - \alpha$ and thus $\mathbb{P}(\eta(\tilde{X}, \tilde{Y}) < -\epsilon \mid \tilde{X}) \leq \mathbb{P}(G(-\epsilon) \geq 1 - \alpha \mid \tilde{X})$ a.s. Moreover, we have

$$\mathbb{P}(G(-\epsilon) \geq 1 - \alpha \mid \tilde{X}) = \mathbb{P}\left(\frac{\sum_{i=1}^n \hat{w}_i \mathbb{1}(V_i \leq -\epsilon)}{\sum_{i=1}^n \hat{w}_i + \hat{w}(\tilde{X}, \tilde{Y})} \geq 1 - \alpha \mid \tilde{X}\right)$$

$$\leq \mathbb{P}\left(\frac{\sum_{i=1}^n \hat{w}_i \mathbb{1}(V_i \leq -\epsilon)}{\sum_{i=1}^n \hat{w}_i} \geq 1 - \alpha \mid \tilde{X}\right) \quad (28)$$

$$= \mathbb{P}\left(\frac{\sum_{i=1}^n \hat{w}_i \mathbb{1}(V_i \leq -\epsilon)}{\sum_{i=1}^n \hat{w}_i} \geq 1 - \alpha\right) \quad (29)$$

where, to get from (28) to (29) we use the independence of $\{(X_i, Y_i)\}_{i=1}^n$ and $\tilde{X}$. Now we observe that

$$\frac{\sum_{i=1}^n \hat{w}_i \mathbb{1}(V_i \leq -\epsilon)}{n} = \frac{\sum_{i=1}^n (\hat{w}_i - w_i)\mathbb{1}(V_i \leq -\epsilon)}{n} + \frac{\sum_{i=1}^n w_i \mathbb{1}(V_i \leq -\epsilon)}{n}.$$

As $n \to \infty$, the strong law of large numbers yields

$$\left|\frac{\sum_{i=1}^n (\hat{w}_i - w_i)\mathbb{1}(V_i \leq -\epsilon)}{n}\right| \xrightarrow{a.s.} \left|\mathbb{E}_{(X,Y)\sim P_{X,Y}^{\pi^b}}\left[(\hat{w}(X,Y) - w(X,Y))\mathbb{1}(s(X,Y) \leq -\epsilon)\right]\right|$$

$$\leq \mathbb{E}_{(X,Y)\sim P_{X,Y}^{\pi^b}}\left[|\hat{w}(X,Y) - w(X,Y)|\mathbb{1}(s(X,Y) \leq -\epsilon)\right]$$

$$\leq \mathbb{E}_{(X,Y)\sim P_{X,Y}^{\pi^b}}\left[|\hat{w}(X,Y) - w(X,Y)|\right] \xrightarrow{m\to\infty} 0 \quad (30)$$

from Assumption 1 and

$$\frac{\sum_{i=1}^n w_i \mathbb{1}(V_i \leq -\epsilon)}{n} \xrightarrow{a.s.} \mathbb{E}_{(X,Y)\sim P_{X,Y}^{\pi^b}}\left[w(X,Y)\mathbb{1}(s(X,Y) \leq -\epsilon)\right] = \mathbb{P}_{(X,Y)\sim P_{X,Y}^{\pi^*}}(s(X,Y) \leq -\epsilon). \quad (31)$$

Using the triangle inequality,

$$\mathbb{P}_{(X,Y)\sim P_{X,Y}^{\pi^*}}(s(X,Y) \leq -\epsilon) \leq \mathbb{P}_{(X,Y)\sim P_{X,Y}^{\pi^*}}(s^*(X,Y) \leq -\epsilon/2) + \mathbb{P}(H(X) \geq \epsilon/2) \quad (32)$$

$$\leq \mathbb{P}_{(X,Y)\sim P_{X,Y}^{\pi^*}}(s^*(X,Y) \leq 0) - \epsilon b_1/2 + 2^k \mathbb{E}[H^k(X)]/\epsilon^k \quad (33)$$

$$= 1 - \alpha - \epsilon b_1/2 + 2^k \mathbb{E}[H^k(X)]/\epsilon^k \xrightarrow{m\to\infty} 1 - \alpha - \epsilon b_1/2.$$

To get from (32) to (33), we use Assumption 2 and Markov's inequality. Similarly, we have

$$\frac{\sum_{i=1}^n \hat{w}_i}{n} = \frac{\sum_{i=1}^n (\hat{w}_i - w_i)}{n} + \frac{\sum_{i=1}^n w_i}{n}$$

so, as $n \to \infty$,

$$\left|\frac{\sum_{i=1}^n (\hat{w}_i - w_i)}{n}\right| \xrightarrow{a.s.} \left|\mathbb{E}_{(X,Y)\sim P_{X,Y}^{\pi^b}}\left[(\hat{w}(X,Y) - w(X,Y))\right]\right|$$

$$\leq \mathbb{E}_{(X,Y)\sim P_{X,Y}^{\pi^b}}\left[|\hat{w}(X,Y) - w(X,Y)|\right] \xrightarrow{m\to\infty} 0, \quad (34)$$

and

$$\frac{\sum_{i=1}^n w_i}{n} \xrightarrow{a.s.} \mathbb{E}_{(X,Y)\sim P_{X,Y}^{\pi^b}}[w(X,Y)] = 1. \quad (35)$$

Putting this all together using the continuous mapping theorem, we get that, almost surely,

$$\lim_{m \to \infty} \lim_{n \to \infty} \frac{\sum_{i=1}^{n} \hat{w}_i \mathbb{1}(V_i \leq -\epsilon)}{\sum_{i=1}^{n} \hat{w}_i} = \lim_{m \to \infty} \lim_{n \to \infty} \frac{\sum_{i=1}^{n} \hat{w}_i \mathbb{1}(V_i \leq -\epsilon)/n}{\sum_{i=1}^{n} \hat{w}_i/n} = 1 - \alpha - \epsilon b_1/2. \quad (36)$$

Since convergence almost surely implies convergence in probability, we have

$$\lim_{m \to \infty} \lim_{n \to \infty} \mathbb{P} \left( \frac{\sum_{i=1}^{n} \hat{w}_i \mathbb{1}(V_i \leq -\epsilon)}{\sum_{i=1}^{n} \hat{w}_i} \geq 1 - \alpha \right) = 0. \quad (37)$$

This implies that, for any $\epsilon > 0$, $\lim_{m \to \infty} \lim_{n \to \infty} \mathbb{P}(\eta(\tilde{X}, \tilde{Y}) < -\epsilon \mid \tilde{X}) = 0$ almost surely.

Using Markov's inequality and Assumption 3

$$\mathbb{P}(H(X) > \epsilon) \leq \mathbb{E}[H^k(X)]/\epsilon^k \overset{m \to \infty}{\longrightarrow} 0. \quad (38)$$

So as $m \to \infty$, $H(X) \overset{\mathcal{P}}{\to} 0$. Similarly, $\mathbb{1}(H(X) > \epsilon) \overset{\mathcal{P}}{\to} 0$ as $m \to \infty$.

Recall (using 27) that, for any $\epsilon \in (0, r/2)$, almost surely,

$$\mathbb{P}(\tilde{Y} \in \hat{C}(\tilde{X}) \mid \tilde{X}) - (1 - \alpha - b_2\epsilon) \geq -b_2 H(\tilde{X}) - \mathbb{1}(H(\tilde{X}) > \epsilon) - \mathbb{P}(\eta(\tilde{X}, \tilde{Y}) < -\epsilon \mid \tilde{X}). \quad (39)$$

For given $t > 0$, pick $\epsilon < \min(r/2, t/2b_2)$. Then,

$$\mathbb{P}(\tilde{Y} \in \hat{C}(\tilde{X}) \mid \tilde{X}) - (1 - \alpha - t/2) \geq -b_2 H(\tilde{X}) - \mathbb{1}(H(\tilde{X}) > \epsilon) - \mathbb{P}(\eta(\tilde{X}, \tilde{Y}) < -\epsilon \mid \tilde{X}). \quad (40)$$

Each term on the right hand side of (40) converges in probability to 0 as $m, n \to \infty$, and therefore using continuous mapping theorem

$$b_2 H(\tilde{X}) + \mathbb{1}(H(\tilde{X}) > \epsilon) + \mathbb{P}(\eta(\tilde{X}, \tilde{Y}) < -\epsilon \mid \tilde{X}) \overset{\mathcal{P}}{\to} 0.$$

This implies

$$\mathbb{P}(\mathbb{P}(\tilde{Y} \in \hat{C}(\tilde{X}) \mid \tilde{X}) \leq 1 - \alpha - t)$$
$$\leq \mathbb{P}(b_2 H(\tilde{X}) + \mathbb{1}(H(\tilde{X}) > \epsilon) + \mathbb{P}(\eta(\tilde{X}, \tilde{Y}) < -\epsilon \mid \tilde{X}) \geq t/2) \to 0.$$

Therefore,

$$\lim_{m \to \infty} \lim_{n \to \infty} \mathbb{P}(\mathbb{P}(\tilde{Y} \in \hat{C}(\tilde{X}) \mid \tilde{X}) \leq 1 - \alpha - t) = 0. \quad (41)$$

$\square$

# B Conformal Off-Policy Prediction (COPP)

## B.1 Further comments on the differences between [14] and COPP

In this subsection, we elaborate on the differences between our work and [14].

Firstly, [14] consider a setup in which the distribution of $X$ is shifted, and construct intervals on the outcome under a specific (deterministic) action, i.e. $Y(a)$. In contrast, we consider a setup in which the distribution of $Y|X$ is shifted due to a change in the policy which is non trivial, and construct bounds on the outcome under this new policy (which could be stochastic). Additionally, since the theory in our methodology relies on the ratio of the joint distribution $P_{X,Y}$, our framework can be straightforwardly extended to the case where both, the conditional $P_{Y|X}$ and the covariate distribution $P_X$ shift.

Secondly, as already mentioned in section 5, [14] can only be applied to the case where we have a deterministic target policy and a discrete action space, whereas COPP generalizes to the stochastic policy and continuous action space. This limitation of [14] can be partially addressed by employing the "*union method*" as described in the main text, which consists of constructing CP intervals for each action separately before taking the union of the intervals. However, we showed in our experiments that this leads to overly conservative intervals i.e. coverage above the required $1 - \alpha$ in Table 1a. This is because the predictive interval does not depend on the target policy, since every action is treated identically when taking the union. This approach is moreover unsuitable for continuous action spaces, whereas COPP applies without modification.

Thirdly, as stated in in section 5, even in the case when we only consider deterministic target policies, there is an important methodological difference between COPP and [14]. [14] construct the intervals on $Y(a)$ by only using calibration data with $A = a$ (see eq. 3.4 in [14]). In contrast, it can be shown that COPP uses the entire calibration data when constructing intervals on $Y(a)$. This is a consequence of integrating out the actions in the weights $w(x, y)$ (sec 3.1). This empirically leads to smaller variance in coverage compared to [14] as evidenced by the experimental results in B.2.

Finally, in our paper we are *not* interested in a linear combination of the $Y(a)$ as in [14], who consider the linear combination of the form $Y(1) - Y(0)$. Instead, as described in section 1.1, we are interested in the outcome $Y$ under the new target policy $\pi^*$ (sometimes denoted as $Y(\pi^*)$ in the literature), which cannot be expressed as a linear combination of $Y(a)$. As a result, there does not appear to be a straightforward application of [14, Section 4.3] to our setup which relies on the linear combination assumption to be applicable.

## B.2 Comparison with [14] on deterministic target policies.

In order to further clarify the distinction between COPP and [14], we conducted additional experiments when the target policy is deterministic i.e. $\pi^*(A|X) = \mathbb{1}\{A = a\}$. In the main text we modified [14] to our setting of stochastic policies by constructing the conformal intervals through the union of the CP sets across the actions. Here we aim to apply COPP to the setting of [14], i.e. deterministic target policy.

As mentioned in in the main text, given that we are integrating out the action in Eq. 7, we are essentially able to use the full dataset when constructing the CP intervals. To see this explicitly, consider the case where $Y \mid X, A$ is a normal random variable (as in our toy experiment). In this case, it can be straightforwardly shown that the weights $w(x_i, y_i)$ will be non-zero, and therefore, when constructing the COPP intervals using (5), we are able to use all the calibration datapoints.

This is contrary to [14], who only consider calibration data with $A = a$, when constructing the CP intervals for $Y(a)$. Below, we use the same experimental setup as our toy experiment in section 6.1 (see section D.1 for more details) with the difference here that we now consider deterministic target policies. In figure 2 we plot the coverage for given deterministic target policies against the number of calibration datapoints. In this figure, we refer to the methodology of [14] as *LC21*. Here, we use the behavioural policy $\pi_{0.3}$ and a deterministic target policy which takes a single fixed action $a \in \{1, 2, 3, 4\}$ at test time. In the title of each subfigure, $Y(a)$ corresponds to the outcome for the target policy $\pi^*(A = a \mid X) = \mathbb{1}(A = a)$.

**Results**: We first note in Figure 2 that the coverage of COPP intervals has a lower variance than [14]. Given that COPP is able to use all the data when constructing the CP intervals, as opposed to [14]

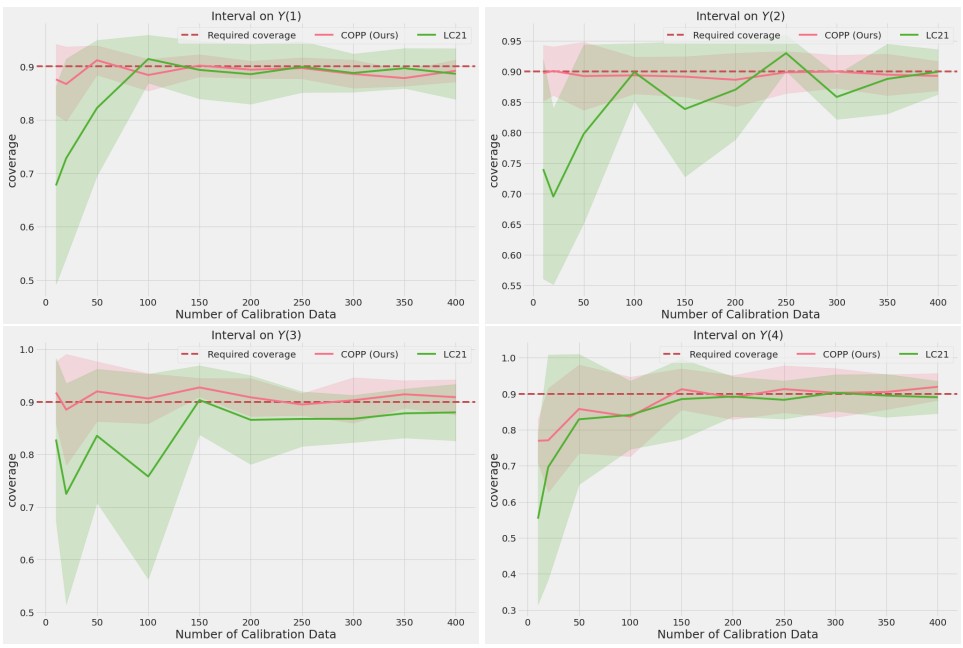

Figure 2: Results for synthetic data experiment with $\pi^b = \pi_{0.3}$ and deterministic target policies.

which only uses a subset, our bounds have lower variance while also attaining the coverage guarantees. We observe this difference particularly in the case when we have little calibration data. Given that [14] have to split the data into 4 different splits (we have 4 different actions), the calibration data for each action is relatively small, whereas we are able to use the whole dataset to construct our CP intervals.

### B.3 Motivation of using stochastic policies for bandits

One of the key difference between our method and that of [14] is that our method can be applied to the setting where the target policy is stochastic. In many settings, deterministic target policies might not be applicable such as in the settings of recommendation systems or RL where exploration is needed [25; 22]. For example, COPP can be used to compare different recommendation systems given some logged data. We explore this application in our MSR experiments where the target policies correspond to different recommendation systems which are, by default, stochastic. Other applications which also make use of stochastic policies bandit problems can be found in [22; 7].

### B.4 COPP for Group-balanced coverage

As [2] point out, we may want predictive intervals that have same error rates across multiple different groups. Using our example of a recommendation system, we may want the predictive intervals to have same coverage across male and female users.

Formally, this problem can be expressed as follows. Let $\Omega = \{\Omega_1, \cdots, \Omega_k\}$ be subsets of $\mathcal{X} \times \mathcal{Y}$ with $\mathbb{P}_{(X,Y) \sim P_{X,Y}^{\pi^*}}((X,Y) \in \Omega_j) > 0$ for $j \in \{1, \ldots, k\}$. We would like to construct predictive intervals $\hat{C}_n^{\Omega}$ which satisfy

$$\mathbb{P}_{(X,Y) \sim P_{X,Y}^{\pi^*}}(Y \in \hat{C}_n^{\Omega}(X) \mid (X,Y) \in \Omega_j) \geq 1 - \alpha \text{ for all } j \in \{1, \ldots, k\}.$$

CP offers us the ability to construct such intervals $\hat{C}_n^{\Omega}$, by simply running algorithm 1 (main text) on each group separately. This has been visualized in figure 3.

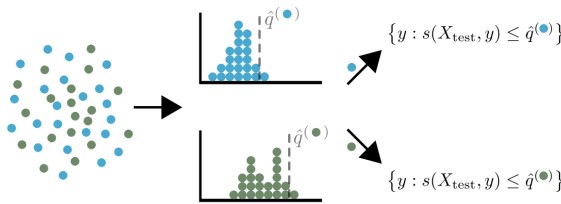

Figure 3: Figure taken from [2]. To achieve group-balanced coverage, we simply run conformal prediction separately on each group.

Formally, this procedure can be described as follows. We group scores into different groups according to each subset.

$$\{(X_i^{\Omega_j}, Y_i^{\Omega_j})\}_{i=1}^{n_j} := \{(X_i, Y_i) : (X_i, Y_i) \in \Omega_j\}_{i=1}^n \text{ and,}$$
$$V_i^{\Omega_j} := (X_i^{\Omega_j}, Y_i^{\Omega_j})$$

Then, within each subset, we calculate the conformal quantile,

$$\eta^{\Omega_j}(x, y) := \text{Quantile}_{1-\alpha}(\hat{F}_n^{\Omega_j}(x, y))$$

where,

$$\hat{F}_n^{\Omega_j}(x, y) := \sum_{i=1}^{n_j} p_i^{\Omega_j}(x, y)\delta_{V_i^{\Omega_j}} + p_{n+1}^{\Omega_j}(x, y)\delta_\infty \text{ where,}$$

$$p_i^{\Omega_j}(x, y) := \frac{w(X_i^{\Omega_j}, Y_i^{\Omega_j})}{\sum_{i=1}^{n_j} w(X_i^{\Omega_j}, Y_i^{\Omega_j}) + w(x, y)}$$

$$p_{n+1}^{\Omega_j}(x, y) := \frac{w(x, y)}{\sum_{i=1}^{n_j} w(X_i^{\Omega_j}, Y_i^{\Omega_j}) + w(x, y)}$$

Next, we construct the set $\hat{C}_n^\Omega$ as follows:

$$\hat{C}_n^\Omega(x^{test}) := \bigcup_{j=1}^k \hat{C}_n^{\Omega_j}(x^{test}) \text{ where,}$$
$$\hat{C}_n^{\Omega_j}(x^{test}) := \{y : (x^{test}, y) \in \Omega_j \text{ and } s(x^{test}, y) \le \eta^{\Omega_j}(x^{test}, y)\}.$$

$$\tag{42}$$

**Proposition B.1** (Coverage guarantee for class-balanced conformal prediction)**.** *Let* $\Omega = \{\Omega_1, \cdots, \Omega_k\}$ *be subsets of* $\mathcal{X} \times \mathcal{Y}$ *with* $\mathbb{P}_{(X,Y) \sim P_{X,Y}^{\pi^*}}((X, Y) \in \Omega_j) > 0$ *for* $j \in \{1, \ldots, k\}$. *Then, the set* $\hat{C}_n^\Omega$ *defined above satisfies the coverage guarantee*

$$\mathbb{P}_{(X,Y) \sim P_{X,Y}^{\pi^*}}(Y \in \hat{C}_n^\Omega(X) \mid (X, Y) \in \Omega_j) \ge 1 - \alpha \text{ for all } j \in \{1, \ldots, k\}.$$

**Proof of Proposition B.1**

$$\mathbb{P}_{(X,Y) \sim P_{X,Y}^{\pi^*}}(Y \in \hat{C}_n^\Omega(X) \mid (X, Y) \in \Omega_j)$$
$$\ge \mathbb{P}_{(X,Y) \sim P_{X,Y}^{\pi^*}}(Y \in \hat{C}_n^{\Omega_j}(X) \mid (X, Y) \in \Omega_j)$$
$$\ge \mathbb{P}_{(X,Y) \sim P_{X,Y}^{\pi^*}}((X, Y) \in \Omega_j : s(X, Y) \le \eta^{\Omega_j}(X, Y) \mid (X, Y) \in \Omega_j) \tag{43}$$

Define the measure $P_{X,Y}^j$ by restricting $P_{X,Y}^{\pi^*}$ to $\Omega_j$, i.e.

$$P_{X,Y}^j(x, y) \propto P_{X,Y}^{\pi^*}(x, y)\mathbb{1}((x, y) \in \Omega_j)$$

Then, (43) can be written as

$$(43) = \mathbb{P}_{(X,Y) \sim P_{X,Y}^j}\left(s(X,Y) \leq \eta^{\Omega_j}(X,Y)\right) \tag{44}$$

Moreover, for $(x,y) \in \Omega_j$ we have

$$w(x,y) = \frac{P_{X,Y}^{\pi^*}(x,y)}{P_{X,Y}^{\pi^b}(x,y)} \propto \frac{P_{X,Y}^j(x,y)}{P_{X,Y}^{\pi^b}(x,y)}$$

Since $p_i^{\Omega_j}(x,y)$ is invariant to scaling of weights $w(x,y)$, replacing the weights by $\tilde{w}(x,y) = \frac{P_{X,Y}^j(x,y)}{P_{X,Y}^{\pi^b}(x,y)}$ keeps the conformal sets unchanged.

Therefore, using Proposition 4.1, the conformal sets constructed will provide coverage guarantees under the measure $P_{X,Y}^j$, i.e.

$$\mathbb{P}_{(X,Y) \sim P_{X,Y}^j}\left(s(X,Y) \leq \eta^{\Omega_j}(X,Y)\right) \geq 1 - \alpha$$

Using (44), we get that

$$\mathbb{P}_{(X,Y) \sim P_{X,Y}^{\pi^*}}\left(Y \in \hat{C}_n^{\Omega}(X) \mid (X,Y) \in \Omega_j\right) \geq \mathbb{P}_{(X,Y) \sim P_{X,Y}^j}\left(s(X,Y) \leq \eta^{\Omega_j}(X,Y)\right) \geq 1 - \alpha$$

$\square$

### B.4.1 COPP for class-balanced coverage

---

**Algorithm 2:** COPP for class-balanced coverage

---

**Inputs:** Observational data $\mathcal{D}_{obs} = \{X_i, A_i, Y_i\}_{i=1}^{n_{obs}}$, conf. level $\alpha$, a score function $s(x,y) \in \mathbb{R}$, new data point $x^{test}$, target policy $\pi^*$ ;

**Output:** $\hat{C}_n^{\mathcal{Y}}(x^{test})$ with coverage guarantee (45);

Split $\mathcal{D}_{obs}$ into training data ($\mathcal{D}_{tr}$) and calibration data ($\mathcal{D}_{cal}$) of sizes $m$ and $n$ respectively;

Use $\mathcal{D}_{tr}$ to estimate weights $\hat{w}(\cdot, \cdot)$;

**for** $y \in \mathcal{Y}$ **do**

    Let $\{X_j^y, Y_j^y\}_{j=1}^{n_y}$ be the following subset of calibration data: $\{(X_i, Y_i) : Y_i = y\}$;

    Let $V_j^y := s(X_j^y, Y_j^y)$, for $j = 1, \ldots, n_y$;

    Define $\hat{F}_n^{x,y} = \sum_{i=1}^{n_y} p_i^w(x,y) \delta_{V_i^y} + p_{n+1}^w(x,y) \delta_\infty$;

    where, $p_i^w(x,y) := \frac{w(X_i^y, Y_i^y)}{\sum_{i=1}^{n_y} w(X_i^y, Y_i^y) + w(x,y)}$, $p_{n+1}^w(x,y) := \frac{w(x,y)}{\sum_{i=1}^{n_y} w(X_i^y, Y_i^y) + w(x,y)}$;

    $\eta(x,y) := \text{Quantile}_{1-\alpha}(\hat{F}_n^{x,y})$

**end**

Define $\hat{C}_n^{\mathcal{Y}}(x^{test}) := \{y : s(x^{test}, y) \leq \eta(x^{test}, y)\}$;

**Return** $\hat{C}_n^{\mathcal{Y}}(x^{test})$

---

In the case when $Y$ is discrete, we construct predictive sets, $\hat{C}_n^{\mathcal{Y}}(x)$, which offer label conditioned coverage guarantees using the methodology described above,

$$\mathbb{P}_{(X,Y) \sim P_{X,Y}^{\pi^*}}\left(Y \in \hat{C}_n^{\mathcal{Y}}(X) \mid Y = y\right) \geq 1 - \alpha, \quad \text{for all } y \in \mathcal{Y} \tag{45}$$

This is a strictly stronger guarantee than marginal coverage, i.e. $\mathbb{P}_{(X,Y) \sim P_{X,Y}^{\pi^*}}(Y \in \hat{C}_n(X)) \geq 1 - \alpha$. To understand what (45) means, consider our running example of recommendation systems, where the outcome $Y$ is whether the recommendation is relevant (0) or not (1) to the user. Then, Eq. (45) ensures that out of the users who received irrelevant recommendations, the predictive sets contain 'not relevant' (1) at least $100 \cdot (1 - \alpha)\%$ of the times. This can be thought of as controlling the false negative rate of irrelevant recommendations at $100 \cdot \alpha\%$. The same is true for users who receive relevant recommendations. This is particularly useful when data is imbalanced, for example when majority of the users in observational receive relevant recommendations.

## B.5  Weights estimation $\hat{w}(x, y)$

### B.5.1  Consistent estimation of the weights does not imply consistent estimation of $\hat{P}(y|x, a)$

In Proposition 4.1, we assume to have consistent estimator of $w(x, y)$ which begs the following question: In general, does a consistent estimate of $w(x, y)$ imply that we also obtain a consistent estimate of $P(y|x, a)$? In particular, one could then just use the estimate of $\hat{P}(y|x, a)$ to construct the predictive interval. However, we answer the above question with the negative by supplying a counter example.

**Counter-example:**

Let $X \in [1, +\infty)$, $a \in \mathbb{R}$ s.t. $|a| < K$ for $K \in \mathbb{R}_{>0}$.

Let $Y|X, a \sim \mathcal{N}((KX^2 + a)^{0.5}, (KX^2 - a))$.

We have $\mathbb{E}[Y^2|X, a] = Var(Y|X, a) + \mathbb{E}[Y|X, a]^2 = KX^2 + a + KX^2 - a = 2KX^2$ (independent of $a$)

Next let

$$\hat{P}(y|x, a) := \frac{y^2 P(y|x, a)}{2Kx^2}. \tag{46}$$

Recall that

$$w(x, y) = \frac{\int P(y|x, a)\pi^*(a|x)\mathrm{d}a}{\int P(y|x, a)\pi^b(a|x)\mathrm{d}a} \tag{47}$$

Using the above definition of $\hat{P}(y|x, a)$ we have:

$$\begin{aligned}
\hat{w}(x, y) &= \frac{\int \hat{P}(y|x, a)\pi^*(a|x)\mathrm{d}a}{\int \hat{P}(y|x, a)\pi^b(a|x)\mathrm{d}a} \\
&= \frac{\int P(y|x, a)\frac{Y^2}{2KX^2}\pi^*(a|x)\mathrm{d}a}{\int P(y|x, a)\frac{Y^2}{2KX^2}\pi^b(a|x)\mathrm{d}a} \\
&= w(x, y).
\end{aligned}$$

Hence, $w(x, y) \equiv \hat{w}(x, y) \implies \hat{P}(y|x, a) \equiv P(y|x, a)$. $\qquad\square$

More generally, if there exists a function $\Phi : \mathcal{X} \times \mathcal{Y} \to \mathbb{R}_{\geq 0}$ such that

1. $\Phi(x, y)$ is not constant in $y$
2. $0 < \mathbb{E}[\Phi(X, Y) \mid X, A] < \infty$, and does not depend on $A$

Then, we can define $\tilde{P}(y|x, a) := P(y|x, a)\Phi(x, y)/\mathbb{E}[\Phi(X, Y) \mid X, A]$, and the weights computed using $\tilde{P}(y|x, a)$ will be the equal to $w(x, y)$ even though $\tilde{P}(y|x, a) \neq P(y|x, a)$.

### B.5.2  Alternative ways to estimate $\hat{w}(x, y)$ without estimating $\hat{P}(y|x, a)$

In this section, we show how we could estimate $w(x, y)$ without having to estimate $\hat{P}(y|x, a)$. One way to obtain an estimate $\hat{w}(x, y)$ is by taking a closer look at the definition of $w(x, y)$ and rewriting the ratio.

$$\begin{aligned}
w(x, y) &= \frac{P_{X,Y}^{\pi^*}(x, y)}{P_{X,Y}^{\pi^b}(x, y)} \\
&= \int \frac{P_{X,A,Y}^{\pi^*}(x, a, y)}{P_{X,A,Y}^{\pi^b}(x, a, y)} P_{A|X,Y}^{\pi^b}(a|x, y)\mathrm{d}a \\
&= \int \frac{\pi^*(a|x)}{\pi^b(a|x)} P_{A|X,Y}^{\pi^b}(a|x, y)\mathrm{d}a \\
&= \mathbb{E}_{A \sim P_{A|X=x, Y=y}^{\pi^b}}\left[\frac{\pi^*(A|x)}{\pi^b(A|x)}\right]. \tag{48}
\end{aligned}$$

**Lemma B.2.** *Let $w(x,y) = \frac{P^{\pi^*}_{X,Y}(x,y)}{P^{\pi^b}_{X,Y}(x,y)}$, then*

$$w(x,y) = \arg\min_f \mathbb{E}_{X,A,Y \sim P^{\pi^b}_{X,A,Y}} \left[ \left\| \frac{\pi^*(A|X)}{\pi^b(A|X)} - f(X,Y) \right\|^2 \right]. \tag{49}$$

**Proof of Lemma B.2**   This follows directly from the identity (48). We prove it here for sake of completeness.

$$\mathbb{E}_{X,A,Y \sim P^{\pi^b}_{X,A,Y}} \left[ \left\| \frac{\pi^*(A|X)}{\pi^b(A|X)} - f(X,Y) \right\|^2 \right]$$

$$= \mathbb{E}_{X,Y \sim P^{\pi^b}_{X,Y}} \left[ \mathbb{E}_{A \sim P^{\pi^b}_{A|X,Y}} \left\| \frac{\pi^*(A|X)}{\pi^b(A|X)} - f(X,Y) \right\|^2 \right]$$

$$= \mathbb{E}_{X,Y \sim P^{\pi^b}_{X,Y}} \left[ \mathrm{Var}_{A \sim P^{\pi^b}_{A|X,Y}} \left[ \frac{\pi^*(A|X)}{\pi^b(A|X)} \right] + \left( \mathbb{E}_{A \sim P^{\pi^b}_{A|X,Y}} \left[ \frac{\pi^*(A|X)}{\pi^b(A|X)} \right] - f(X,Y) \right)^2 \right]. \tag{50}$$

Where, (50) is minimized if $f(x,y) = \mathbb{E}_{A \sim P^{\pi^b}_{A|X=x,Y=y}} \left[ \frac{\pi^*(A|x)}{\pi^b(A|x)} \right] = w(x,y)$.   $\square$

Using Lemma B.2, we can thus approximate $w(x,y)$ by minimizing the loss

$$\hat{w}(x,y) = \arg\min_{f_\theta} \mathbb{E}_{X,A,Y \sim P^{\pi^b}_{X,A,Y}} \left[ \left\| \frac{\pi^*(A|X)}{\pi^b(A|X)} - f_\theta(X,Y) \right\|^2 \right] \tag{51}$$

Hence we see that the ratio estimation problem can be rewritten as a regression problem where $f_\theta(x,y)$ is for example a neural network. This allows one to estimate directly, without the need for estimating $\hat{P}(y \mid x, a)$ first.

# C Estimation of the quantiles of the target distribution

As mentioned in Section 4.2, we present here a way to estimate the quantiles of the target distribution $P_{X,Y}^{\pi^*}$ consistently when the ground truth weight function $w(x,y)$ is known. As we are interested in the quantiles, we will be using the pinball loss to train our model $\hat{f}_\theta$ defined by

$$
L_\alpha(\theta, x, y) = \begin{cases} \alpha(\hat{f}_\theta(x) - y) & \text{if } (\hat{f}_\theta(x) - y) > 0, \\ (1 - \alpha)(y - \hat{f}_\theta(x)) & \text{if } (\hat{f}_\theta(x) - y) < 0. \end{cases}
$$

Then we have the following objective to optimize:

$$
\begin{aligned}
\mathbb{E}_{(X,Y) \sim P_{X,Y}^{\pi^*}}[L_\alpha(\theta, X, Y)] &= \int_{X,Y} L_\alpha(\theta, x, y) P_{X,Y}^{\pi^*}(\mathrm{d}x, \mathrm{d}y) \\
&= \int_{X,Y} L_\alpha(\theta, x, y) \frac{\mathrm{d}P_{X,Y}^{\pi^*}(x, y)}{\mathrm{d}P_{X,Y}^{\pi^b}(x, y)} P_{X,Y}^{\pi^b}(\mathrm{d}x, \mathrm{d}y) \\
&= \int_{X,Y} L_\alpha(\theta, x, y) w(x, y) P_{X,Y}^{\pi^b}(\mathrm{d}x, \mathrm{d}y) \\
&= \mathbb{E}_{(X,Y) \sim P_{X,Y}^{\pi^b}}[L_\alpha(\theta, X, Y) w(X, Y)].
\end{aligned}
$$

The above holds true if the true weight function is known. However in the case where we only have a consistent estimator of $w(x, y)$, it remains to be proven that the above objective will also yield a consistent estimator of the quantiles under $\pi^*$. We leave this for future work to prove as we are simply providing a possible avenue to relax the assumptions in Proposition 4.2.

# D    Experiments

The code for our experiments is available at `https://anonymous.4open.science/r/COPP-75F5` and we ran all our experiments on Intel(R) Xeon(R) CPU E5-2690 v4 @ 2.60GHz with 8GB RAM per core. We were able to use 100 CPUs in parallel to iterate over different configurations and seeds. However, we would like to note that our algorithms only requires 1 CPU and at most 10 mins to run, as our networks are relatively small.

## D.1    Toy Experiment

### D.1.1    Synthetic data experiments setup

**Model.**    The observational data distribution is defined as follows:

$$X_i \overset{\text{i.i.d.}}{\sim} \mathcal{N}(0, 9)$$
$$A_i \mid x_i \sim \pi^b(\cdot \mid x_i) \ \text{ where } \pi^b \text{ has been defined below}$$
$$Y_i \mid x_i, a_i \sim \mathcal{N}(a_i * x_i, 1)$$

**Behaviour and Target Policies.**    We define a family of policies $\pi_\epsilon(a \mid x)$ as follows:

$$\pi_\epsilon(a|x) := \begin{cases} \epsilon \mathbb{1}(a \in \{1, 2, 3\}) + (1 - 3\epsilon)\mathbb{1}(a = 4) & \text{if } |x| \in (3, \infty) \\ \epsilon \mathbb{1}(a \in \{1, 2, 4\}) + (1 - 3\epsilon)\mathbb{1}(a = 3) & \text{if } |x| \in (2, 3] \\ \epsilon \mathbb{1}(a \in \{1, 3, 4\}) + (1 - 3\epsilon)\mathbb{1}(a = 2) & \text{if } |x| \in (1, 2] \\ \epsilon \mathbb{1}(a \in \{2, 3, 4\}) + (1 - 3\epsilon)\mathbb{1}(a = 1) & \text{if } |x| \in [0, 1] \end{cases}$$

We use the parameter $\epsilon \in (0, 1/3)$ to control the policy shift between target and behaviour policies. For the behaviour policy $\pi^b$, we use $\epsilon^b = 0.3$, and for target policies $\pi^*$, we use $\epsilon^* \in \{0.1, 0.2, 0.3\}$. Here we use $m = 1000$ training datapoints.

**Neural Network Architectures**

- To approximate the behaviour policy $\pi^b$, we use a neural network with 2 hidden layers and 16 nodes in each hidden layer, and ReLU activation function.

- To approximate $P(y|x, a)$, we use $\mathcal{N}(\mu(x, a), \sigma(x, a))$, where $\mu$ and $\sigma$ are neural networks with one-hidden layer, 32 nodes in the hidden layer, and ReLU activation function.

- For the score function, we train the quantiles $\hat{q}_{\alpha/2}$ and $\hat{q}_{1-\alpha/2}$ using quantile regression, each of which are modelled using neural networks with one-hidden layer, 32 nodes in the hidden layer, and ReLU activation functions.

**Results: Coverage as a function of increase calibration data**    As mentioned in the main text, we have also performed experiments to investigate how much calibration data is needed for COPP as well as other methods to converge to the required $90\%$ coverage. In the below figure 4 we have plotted the coverage as a function of $n$ calibration data points. Our proposed method is converging much faster to the required coverage compared to the competing methods.

**Additional experimental baseline using weighted quantile regression.**    In order to add an additional baseline that is also covariate dependent, we have added some experiments using the weighted quantile regression (WQR) as described in Sec. C on our toy experiments from Sec. 6 in the main text. Below in Table 3 and Table 4 we see the complete coverage table with the respective interval lengths. Note also that WQR does not seem to perform well as it does not have any statistical guarantees and heavily relies on good estimation of the ratio. We have added these experiments here in the appendix for completeness and did not add it in the main text as the results were not comparable to other baselines.

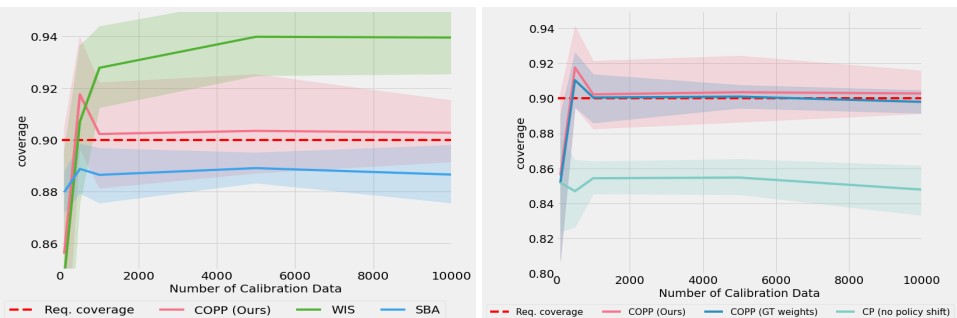

Figure 4: Results for synthetic data experiment with $\pi^b = \pi_{0.3}$ and the target policy is $\pi^* = \pi_{0.1}$. **Left:** our proposed method is able to converge to the required coverage rather quickly compared to the competing methods. **Right:** here we see that our method is on par with using the GT weights. Due to estimation error, COPP with estimated weights has slightly higher variance in terms of coverage

Table 3: Mean Coverage as a function of policy shift with 2 standard errors over 10 runs. We have added weighted quantile regression (WQR) for completeness and note that it does not seem to perform well.

| Coverage | $\Delta_\epsilon = 0.0$ | $\Delta_\epsilon = 0.1$ | $\Delta_\epsilon = 0.2$ |
|---|---|---|---|
| COPP (Ours) | **0.90 ± 0.01** | **0.90 ± 0.01** | **0.91 ± 0.01** |
| WIS | **0.89 ± 0.01** | **0.91 ± 0.02** | 0.94 ± 0.02 |
| SBA | **0.90 ± 0.01** | 0.88 ± 0.01 | 0.87 ± 0.01 |
| COPP (GT weights Ours) | **0.90 ± 0.01** | **0.90 ± 0.01** | **0.90 ± 0.01** |
| CP (no policy shift) | **0.90 ± 0.01** | 0.87 ± 0.01 | 0.85 ± 0.01 |
| CP (union) | 0.96 ± 0.01 | 0.96 ± 0.01 | 0.96 ± 0.01 |
| WQR | 0.82 ± 0.04 | 0.76 ± 0.03 | 0.70 ± 0.03 |

Table 4: Mean Interval Length as a function of policy shift with 2 standard errors over 10 runs. We have added weighted quantile regression (WQR) for completeness and note that it does not seem to perform well.

| Interval Lengths | $\Delta_\epsilon = 0.0$ | $\Delta_\epsilon = 0.1$ | $\Delta_\epsilon = 0.2$ |
|---|---|---|---|
| COPP (Ours) | 9.08 ± 0.10 | 9.48 ± 0.22 | 9.97 ± 0.38 |
| WIS | 24.14 ± 0.30 | 32.96 ± 1.80 | 43.12 ± 3.49 |
| SBA | 8.78 ± 0.12 | 8.94 ± 0.10 | 8.33 ± 0.09 |
| COPP (GT weights Ours) | 8.91 ± 0.09 | 9.25 ± 0.12 | 9.59 ± 0.20 |
| CP (no policy shift) | 9.00 ± 0.10 | 9.00 ± 0.10 | 9.00 ± 0.10 |
| CP (union) | 10.66 ± 0.18 | 11.04 ± 0.2 | 11.4 ± 0.26 |
| WQR | 8.55 ± 0.50 | 8.61 ± 0.52 | 8.70 ± 0.55 |

### D.1.2 Experiments with continuous action space

As mentioned in the main text and also in Sec. B.1, our proposed method, contrary to the work of [14] is able to also handle continuous action space. Given that we are integrating out the actions when computing the weights in Eq. 7 our method trivially extends to the continuous action space, whereas [14] is only applicable for discrete action spaces, as they compute conformal intervals conditioned on a given action.

**Model.** The observational data distribution is defined as follows:

$$X_i \overset{\text{i.i.d.}}{\sim} \mathcal{N}(0, 4)$$
$$A_i \mid x_i \sim \mathcal{N}(x_i/4, 1)$$
$$Y_i \mid x_i, a_i \sim \mathcal{N}(a_i + x_i, 1)$$

**Target Policies.** We define a family of policies $\pi_\epsilon(a \mid x)$ as follows:

$$\pi_\epsilon(a \mid x) = \mathcal{N}(x/4 + \epsilon, 1). \tag{52}$$

In our experiments, for the target policy $\pi^*$, we use $\pi^* = \pi_{\epsilon^*}$ for $\epsilon^* \in \{0, 0.5, 1, 1.5, 2, 2.5\}$.

**Results.** Table 5 shows the coverages of different methods as the policy shift $\epsilon^*$ increases. The behaviour policy $\pi^b = \pi_0$ is fixed and we use $n = 5000$ calibration datapoints and $m = 1000$ training points, across 10 runs. Table 5 shows, how COPP stays very close to the required coverage of $90\%$ across all target policies with $\epsilon^* \leq 2.0$, compared to WIS and SBA. Both, WIS intervals and SBA intervals suffer from under-coverage i.e. below the required coverage. These results again support our hypothesis from Sec. 3.1, which stated that COPP is less sensitive to estimation errors of $\hat{P}(y|x, a)$ compared to directly using $\hat{P}(y|x, a)$ for the intervals i.e. SBA.

Next, Table 6 shows the mean interval lengths and even though WIS intervals are under-covered, the average interval length is huge compared to COPP. Additionally, for $\epsilon^* \in \{0, 0.5, 1, 1.5\}$, COPP with estimated weights produces results which are close to COPP intervals with ground truth weights. This shows that when the behaviour and target policies have reasonable overlap, the effect of weights estimation error on COPP results is limited. However, as $\epsilon^*$ increases to $2.0$ and $2.5$, the overlap between behaviour and target policies becomes low. We empirically note that this leads to high weights estimation error and consequently under-coverage in COPP with estimated weights. In contrast, COPP with ground truth weights still achieves required coverage, even though it becomes conservative when the overlap is low. Figure 5 visualises how the overlap between target and behaviour policies decreases with increasing $\epsilon^*$. It can be seen that $\epsilon^* \in \{2, 2.5\}$ leads to very low overlap between the behaviour and target data.

Table 5: Mean Coverage as a function of policy shift with 2 standard errors over 10 runs.

| Coverage | $\epsilon^* = 0.0$ | $\epsilon^* = 0.5$ | $\epsilon^* = 1.0$ | $\epsilon^* = 1.5$ | $\epsilon^* = 2.0$ | $\epsilon^* = 2.5$ |
|---|---|---|---|---|---|---|
| COPP (Ours) | **0.90 ± 0.01** | **0.91 ± 0.01** | 0.92 ± 0.01 | **0.91 ± 0.01** | **0.89 ± 0.02** | 0.85 ± 0.02 |
| WIS | 0.87 ± 0.01 | 0.87 ± 0.01 | 0.87 ± 0.01 | 0.87 ± 0.02 | **0.89 ± 0.02** | 0.83 ± 0.02 |
| SBA | 0.86 ± 0.01 | 0.86 ± 0.01 | 0.86 ± 0.01 | 0.86 ± 0.01 | **0.89 ± 0.02** | 0.83 ± 0.02 |
| COPP (GT Weights Ours) | **0.90 ± 0.01** | **0.91 ± 0.01** | **0.91 ± 0.01** | **0.90 ± 0.01** | 0.96 ± 0.02 | 0.93 ± 0.02 |
| CP (no policy shift) | **0.90 ± 0.01** | 0.88 ± 0.01 | 0.82 ± 0.01 | 0.73 ± 0.01 | 0.60 ± 0.01 | 0.46 ± 0.01 |

Table 6: Mean Interval Length as a function of policy shift with 2 standard errors over 10 runs.

| Interval Lengths | $\epsilon^* = 0.0$ | $\epsilon^* = 0.5$ | $\epsilon^* = 1.0$ | $\epsilon^* = 1.5$ | $\epsilon^* = 2.0$ | $\epsilon^* = 2.5$ |
|---|---|---|---|---|---|---|
| COPP (Ours) | 4.75 ± 0.04 | 5.08 ± 0.09 | 5.89 ± 0.14 | 6.92 ± 0.18 | 7.82 ± 0.41 | 8.45 ± 0.44 |
| WIS | 9.55 ± 0.1 | 9.56 ± 0.12 | 9.56 ± 0.27 | 9.44 ± 0.38 | 9.40 ± 0.59 | 9.08 ± 0.64 |
| SBA | 4.38 ± 0.03 | 4.37 ± 0.03 | 4.36 ± 0.04 | 4.34 ± 0.07 | 4.31 ± 0.1 | 4.28 ± 0.14 |
| COPP (GT Weights Ours) | 4.73 ± 0.05 | 5.07 ± 0.09 | 5.87 ± 0.14 | 6.82 ± 0.13 | 7.57 ± 0.19 | 8.07 ± 0.22 |
| CP (no policy shift) | 4.70 ± 0.05 | 4.70 ± 0.05 | 4.70 ± 0.05 | 4.70 ± 0.05 | 4.70 ± 0.05 | 4.70 ± 0.05 |

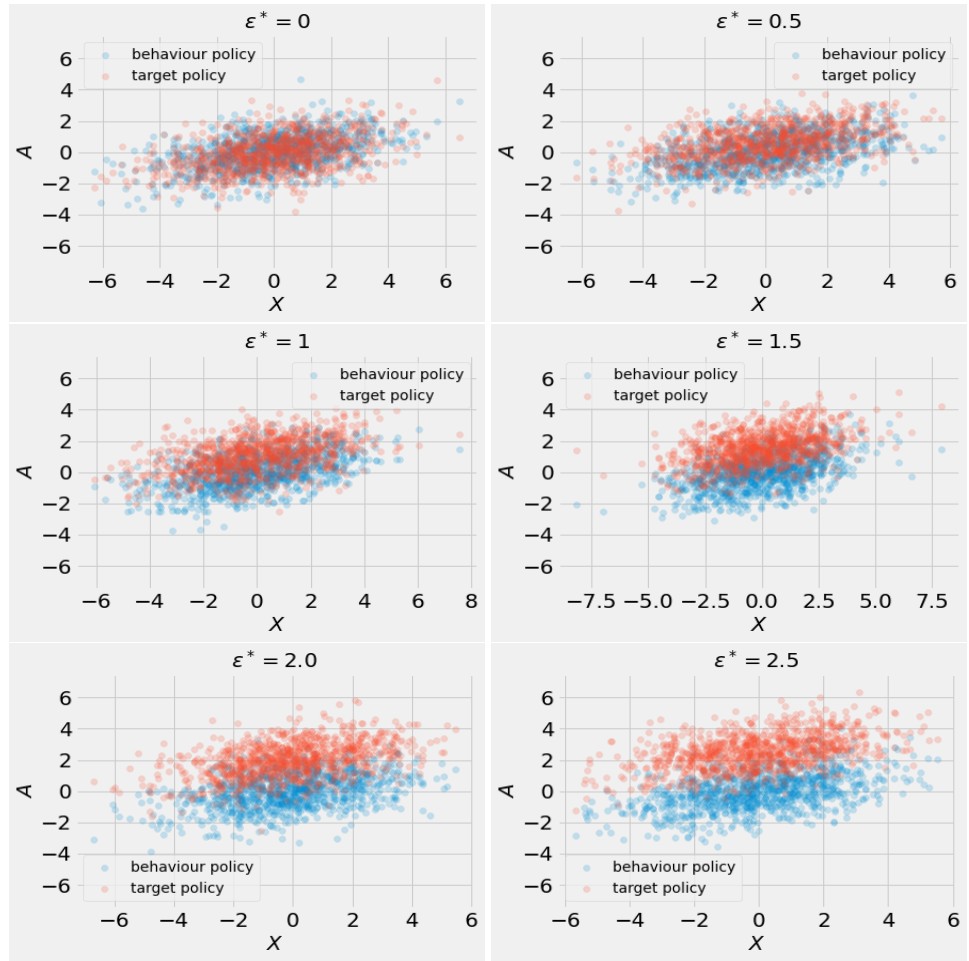

Figure 5: Plots of $A$ against $X$, where $X \sim \mathcal{N}(0, 4)$ and $A \mid X$ is sampled from behaviour and target policies. Here, target policies are defined in (52) for $\epsilon^* \in \{0, 0.5, 1, 1.5, 2, 2.5\}$.

## D.2  Experiments on Microsoft Ranking Dataset

**Dataset details.**  The dataset contains relevance scores for websites recommended to different users, and comprises of $30{,}000$ user-website pairs. For a user $i$ and website $j$, the data contains a 136-dimensional feature vector $u_i^j$, which consists of user $i$'s attributes corresponding to website $j$, such as length of stay or number of clicks on the website. Furthermore, for each user-website pair, the dataset also contains a relevance score, i.e. how relevant the website was to the user.

First, given a user $i$ we sample (with replacement) 5 websites, $\{u_i^j\}_{j=1}^5$, from the data. Next, we reformulate this into a contextual bandit where $A \in \{1, 2, 3, 4, 5\}$ corresponds to the website we recommend to a user. For a user $i$, we define $X$ by combining the 5 feature vectors corresponding to the user, i.e. $X \in \mathbb{R}^{5 \times 136}$, where $x_i = (u_i^1, u_i^2, u_i^3, u_i^4, u_i^5)$. In addition, $Y \in \{0, 1, 2, 3, 4\}$ corresponds to the relevance score for the $A$'th website, i.e. the recommended website. The goal is to construct prediction sets that are guaranteed to contain the true relevance score with a probability of $90\%$. Here we use $m = 5000$ training data points.

**Behaviour and Target Policies.**  We first train a Neural Network (NN) classifier model mapping each 136-dimensional feature vector to the softmax scores for each relevance score class, $\hat{f}_\theta : \mathcal{U} \to [0, 1]^5$. We use this trained model $\hat{f}_\theta$ to define a family of policies such that we pick the most relevant website as predicted by $\hat{f}_\theta$ with probability $\epsilon$ and the rest uniformly with probability $(1 - \epsilon)/4$.

Formally, this has been expressed as follows. We use $\hat{f}_\theta^{\text{label}}$ to denote the relevance class predicted by $\hat{f}_\theta$, i.e. $\hat{f}_\theta^{\text{label}}(u) := \arg\max_i\{\hat{f}_\theta(u)_i\}$.

Then,

$$\pi_\epsilon(a \mid X = (u^1, u^2, u^3, u^4, u^5)) := \epsilon\mathbb{1}(a = \arg\max_j\{\hat{f}_\theta^{\text{label}}(u^j)\})$$
$$+ (1-\epsilon)/4\mathbb{1}(a \neq \arg\max_j\{\hat{f}_\theta^{\text{label}}(u^j)\})$$

**Estimation of ratios, $\hat{w}(X, Y)$.**    To estimate the $\hat{P}(y \mid x, a)$ we use the trained model $\hat{f}_\theta$ as follows:

$$\hat{P}(y \mid x = (u^1, u^2, u^3, u^4, u^5), a) = \hat{f}_\theta(u^a)_y$$

where $\hat{f}_\theta(u^a)_y$ corresponds to the softmax prediction of $u^a$ for label $y$ under the model $\hat{f}_\theta$. To estimate the behaviour policy $\hat{\pi}^b$, we train a classifier model $\mathcal{X} \to \mathcal{A}$ using a neural network. We use (7) to estimate the weights $\hat{w}(x, y)$.

#### Neural Network Architectures

- To approximate the behaviour policy, we use a neural network with 2 hidden layers and 25 nodes in each hidden layer, ReLU activations and softmax output.

- To approximate $\hat{f}_\theta$, we use a neural network with 2 hidden layers with 64 nodes each and ReLU activations.

**Results: Coverage as a function of increase calibration data.**    As mentioned in the main text, we have also performed experiments to investigate how much calibration data is needed for COPP as well as other methods to converge to the required $90\%$ coverage. In the below plot we have plotted the coverage as a function of $n$ calibration data points. We observe that our proposed method is converging much faster to the required coverage compared to the competing methods.

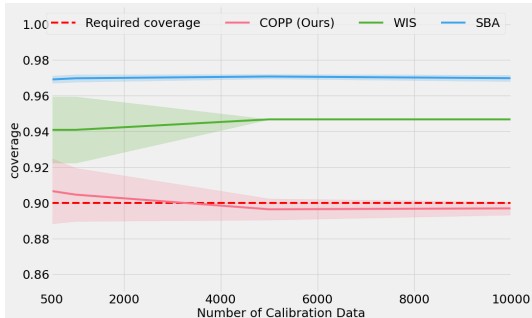

Figure 6: Results of Microsoft Ranking Dataset experiment with behaviour policy $\pi^b = \pi_{0.5}$ and the target policy is $\pi^* = \pi_{0.2}$. Our proposed method is able to converge to the required coverage rather quickly compared to the competing methods

Table 7: Coverages for COPP with and without label conditioned coverage, $\hat{C}_n^{\mathcal{Y}}$ and $\hat{C}_n$ respectively. Overall coverage refers to marginal coverage while $Y = y$ refers to coverage conditioned on $Y = y$. Here $n_{test}$ corresponds to the number of test data points ($\sim P^{\pi^*}$).

|         | $n_{test}$ | $\hat{C}_n$ Cov | $\hat{C}_n^{\mathcal{Y}}$ Cov |
|---------|-----------|------------------|-------------------------------|
| Overall | 5000      | $0.896 \pm 0.005$ | $0.941 \pm 0.003$            |
| $Y = 0$ | 266       | $0.700 \pm 0.020$ | $1.000 \pm 0.000$            |
| $Y = 1$ | 293       | $0.526 \pm 0.019$ | $1.000 \pm 0.000$            |
| $Y = 2$ | 228       | $0.772 \pm 0.018$ | $0.990 \pm 0.029$            |
| $Y = 3$ | 320       | $0.852 \pm 0.015$ | $0.964 \pm 0.035$            |
| $Y = 4$ | 3893      | $0.950 \pm 0.006$ | $0.928 \pm 0.003$            |

### D.2.1 Results: COPP for Class-balanced coverage

Table 7 shows the coverages of COPP predictive sets ($\hat{C}_n$ with marginal coverage guarantee constructed using algorithm 1) and COPP intervals with label conditioned coverage ($\hat{C}_n^{\mathcal{Y}}$ satisfying (45) constructed using algorithm 2). Extensions of WIS and SBA to the conditional case are not straightforward and hence have not been included. For $\hat{C}_n$, while the overall coverage is very close to the required coverage of $90\%$, we see that there is under-coverage for $Y = 0, 1, 2, 3$. This can be explained by the data imbalance – the number of test data points with $Y = 0, 1, 2, 3$ is significantly lower than $Y = 4$.

This under-coverage problem disappears in $\hat{C}_n^{\mathcal{Y}}$. Instead, in cases where number of data points is small, ($Y = 0, 1, 2, 3$), the predictive sets $\hat{C}_n^{\mathcal{Y}}$ are conservative (i.e. have coverage $> 90\%$). As a result, the overall coverage increases to 0.941. This is a price to be paid for label conditioned coverage – the overall coverage may increase, however, being conservative in safety-critical settings is better than being overly optimistic.

### D.3 UCI Dataset experiments

Following [9; 6; 30] we apply COPP on UCI classification datasets. We can pose classification as contextual bandits by defining the covariates $\mathcal{X}$ as the features, the action space $\mathcal{A} = \mathcal{K}$, where $\mathcal{K}$ is the set of labels, and the outcomes are binary, i.e. $\mathcal{Y} = \{0,1\}$, defined by $Y \mid X, A = \mathbb{1}(X \text{ belongs to class } A)$. Here we use $m = 1000$ training data points.

**Behaviour and Target Policies.** First we train a neural network classifier mapping each covariate to the softmax scores for each class, $\hat{f}_\theta : \mathcal{X} \to [0,1]^{|\mathcal{K}|}$. We use this trained model $\hat{f}_\theta$ to define a family of policies such that we pick the most likely label as predicted by $\hat{f}_\theta$ with probability $\epsilon$ and the rest uniformly with probability. Formally, this can be expressed as follows:

$$\pi_\epsilon(a \mid x) := \epsilon \mathbb{1}(a = \arg\max_{k \in \mathcal{K}}\{\hat{f}_\theta(x)_k\}) + (1 - \epsilon)/(|\mathcal{K}| - 1)\mathbb{1}(a \neq \arg\max_{k \in \mathcal{K}}\{\hat{f}_\theta(x)_k\})$$

Like other experiments, we use $\epsilon$ to control the shift between behaviour and target policies. For $\pi^b$, we use $\epsilon^b = 0.5$ and for $\epsilon^* \in \{0.05, 0.3, 0.4, 0.5, 0.6, 0.7, 0.95\}$. Using this behaviour policy $\pi^b$, we generate an observational dataset $\mathcal{D}_{obs} = \{x_i, a_i, y_i\}_{i=1}^{n_{obs}}$ which is then split into training $\mathcal{D}_{tr}$ and calibration datasets $\mathcal{D}_{cal}$, of sizes $m$ and $n$ respectively.

**Estimation of ratios, $\hat{w}(X, Y)$.** To estimate the $\hat{P}(y \mid x, a)$ we use the trained model $\hat{f}_\theta$ as follows:

$$\hat{P}(Y = 1 \mid x, a) = \hat{f}_\theta(x)_a$$

where $\hat{f}_\theta(x)_a$ corresponds the softmax prediction of $x$ for label $a$ under the model $\hat{f}_\theta$. To estimate the behaviour policy $\hat{\pi}^b$, we train a classifier model $\mathcal{X} \to \mathcal{A}$ using a neural network. We use (7) in main text to estimate weights $\hat{w}(x, y)$.

**Score.** We define $\hat{P}^{\pi^b}(y \mid x) = \sum_{i \in \mathcal{K}} \hat{\pi}^b(A = i|x)\hat{P}(y|x, A = i)$. Using similar formulation as in [2], we define the score as

$$s(x, y) = \sum_{y'=0,1} \hat{P}^{\pi^b}(y' \mid x)\mathbb{1}(\hat{P}^{\pi^b}(y' \mid x) \geq \hat{P}^{\pi^b}(y \mid x))$$

**Neural Network Architectures**

- To approximate the behaviour policy, we use a neural network with 2 hidden layers and 64 nodes in each hidden layer, ReLU activations and softmax output.

- To approximate $\hat{f}_\theta$, we use a neural network with 2 hidden layers with 64 nodes each and ReLU activations.

**Results.** Tables 8-13 show the coverages across varying target policies for different classification datasets. The behaviour policy $\pi^b = \pi_{0.5}$ is fixed and we use $n = 5000$ calibration datapoints, across 10 runs with $m = 5000$ training data. The tables show that COPP is able to provide the required coverage of 90% across all target policies. Moreover, compared to COPP, SBA and WIS are overly conservative. WIS estimates are not adaptive w.r.t. $X$, and as a result, the predictive sets produced are uninformative (i.e. contain all outcomes) in these experiments where the outcome is binary.

We have also included a comparison of COPP using estimated behaviour policy with COPP using GT behaviour policy. The latter provides more accurate coverage, and using estimated behaviour policy provides slightly over-covered predictive sets comparatively in most cases. This can be explained by policy estimation error. Additionally, we observe that using standard CP leads to predictive sets which are not adaptive to policy shift. As a result, the standard CP predictive sets get overly conservative (optimistic) as $\Delta_\epsilon$ becomes more negative (positive).

### Table 8: Yeast dataset results

|  | $\Delta_\epsilon = -0.45$ | $\Delta_\epsilon = -0.2$ | $\Delta_\epsilon = -0.1$ | $\Delta_\epsilon = 0.0$ | $\Delta_\epsilon = 0.1$ | $\Delta_\epsilon = 0.2$ | $\Delta_\epsilon = 0.45$ |
|---|---|---|---|---|---|---|---|
| COPP (Ours) | 0.92±0.00 | 0.92±0.00 | 0.92±0.00 | 0.92±0.00 | 0.92±0.00 | 0.92±0.00 | 0.91±0.00 |
| WIS | 0.99±0.01 | 1.00±0.00 | 1.00±0.00 | 1.00±0.00 | 1.00±0.00 | 1.00±0.00 | 1.00±0.00 |
| SBA | 0.98±0.00 | 1.00±0.00 | 1.00±0.00 | 1.00±0.00 | 1.00±0.00 | 1.00±0.00 | 1.00±0.00 |
| COPP (GT behav policy) | 0.91±0.00 | 0.91±0.00 | 0.90±0.00 | 0.90±0.00 | 0.90±0.00 | 0.90±0.00 | 0.90±0.00 |
| CP (no policy shift) | 0.97±0.00 | 0.93±0.00 | 0.92±0.00 | 0.90±0.00 | 0.89±0.00 | 0.87±0.00 | 0.83±0.00 |

### Table 9: Ecoli dataset results

|  | $\Delta_\epsilon = -0.45$ | $\Delta_\epsilon = -0.2$ | $\Delta_\epsilon = -0.1$ | $\Delta_\epsilon = 0.0$ | $\Delta_\epsilon = 0.1$ | $\Delta_\epsilon = 0.2$ | $\Delta_\epsilon = 0.45$ |
|---|---|---|---|---|---|---|---|
| COPP (Ours) | 0.92±0.00 | 0.91±0.00 | 0.91±0.00 | 0.90±0.00 | 0.90±0.00 | 0.90±0.00 | 0.90±0.00 |
| WIS | 1.00±0.00 | 1.00±0.00 | 1.00±0.00 | 1.00±0.00 | 1.00±0.00 | 1.00±0.00 | 1.00±0.00 |
| SBA | 1.00±0.00 | 1.00±0.00 | 1.00±0.00 | 1.00±0.00 | 1.00±0.00 | 1.00±0.00 | 1.00±0.00 |
| COPP (GT behav policy) | 0.91±0.00 | 0.90±0.00 | 0.90±0.00 | 0.90±0.00 | 0.90±0.00 | 0.90±0.00 | 0.90±0.01 |
| CP (no policy shift) | 0.92±0.00 | 0.91±0.00 | 0.91±0.00 | 0.90±0.00 | 0.90±0.00 | 0.89±0.00 | 0.88±0.00 |

### Table 10: Letter dataset results

|  | $\Delta_\epsilon = -0.45$ | $\Delta_\epsilon = -0.2$ | $\Delta_\epsilon = -0.1$ | $\Delta_\epsilon = 0.0$ | $\Delta_\epsilon = 0.1$ | $\Delta_\epsilon = 0.2$ | $\Delta_\epsilon = 0.45$ |
|---|---|---|---|---|---|---|---|
| COPP (Ours) | 0.95±0.00 | 0.93±0.00 | 0.93±0.00 | 0.92±0.00 | 0.92±0.00 | 0.92±0.00 | 0.91±0.00 |
| WIS | 1.00±0.00 | 1.00±0.00 | 1.00±0.00 | 1.00±0.00 | 1.00±0.00 | 1.00±0.00 | 1.00±0.00 |
| SBA | 0.97±0.00 | 1.00±0.00 | 1.00±0.00 | 1.00±0.00 | 1.00±0.00 | 1.00±0.00 | 1.00±0.00 |
| COPP (GT behav policy) | 0.92±0.00 | 0.91±0.00 | 0.91±0.00 | 0.90±0.00 | 0.89±0.00 | 0.89±0.00 | 0.88±0.00 |
| CP (no policy shift) | 0.99±0.00 | 0.94±0.00 | 0.92±0.00 | 0.90±0.00 | 0.88±0.00 | 0.86±0.00 | 0.81±0.00 |

### Table 11: Optdigits dataset results

|  | $\Delta_\epsilon = -0.45$ | $\Delta_\epsilon = -0.2$ | $\Delta_\epsilon = -0.1$ | $\Delta_\epsilon = 0.0$ | $\Delta_\epsilon = 0.1$ | $\Delta_\epsilon = 0.2$ | $\Delta_\epsilon = 0.45$ |
|---|---|---|---|---|---|---|---|
| COPP (Ours) | 0.93±0.00 | 0.93±0.00 | 0.93±0.00 | 0.93±0.00 | 0.93±0.00 | 0.93±0.00 | 0.93±0.00 |
| WIS | 0.99±0.01 | 1.00±0.00 | 1.00±0.00 | 1.00±0.00 | 1.00±0.00 | 1.00±0.00 | 1.00±0.00 |
| SBA | 0.97±0.00 | 1.00±0.00 | 1.00±0.00 | 1.00±0.00 | 1.00±0.00 | 1.00±0.00 | 0.99±0.00 |
| COPP (GT behav policy) | 0.91±0.00 | 0.90±0.00 | 0.90±0.00 | 0.90±0.00 | 0.90±0.00 | 0.89±0.00 | 0.89±0.00 |
| CP (no policy shift) | 0.97±0.00 | 0.93±0.00 | 0.91±0.00 | 0.90±0.00 | 0.88±0.00 | 0.87±0.00 | 0.83±0.00 |

### Table 12: Pendigits dataset results

|  | $\Delta_\epsilon = -0.45$ | $\Delta_\epsilon = -0.2$ | $\Delta_\epsilon = -0.1$ | $\Delta_\epsilon = 0.0$ | $\Delta_\epsilon = 0.1$ | $\Delta_\epsilon = 0.2$ | $\Delta_\epsilon = 0.45$ |
|---|---|---|---|---|---|---|---|
| COPP (Ours) | 0.92±0.00 | 0.92±0.00 | 0.92±0.00 | 0.92±0.00 | 0.92±0.00 | 0.92±0.00 | 0.91±0.00 |
| WIS | 1.00±0.00 | 1.00±0.00 | 1.00±0.00 | 1.00±0.00 | 1.00±0.00 | 1.00±0.00 | 1.00±0.00 |
| SBA | 0.97±0.00 | 1.00±0.00 | 1.00±0.00 | 1.00±0.00 | 1.00±0.00 | 1.00±0.00 | 0.99±0.00 |
| COPP (GT behav policy) | 0.91±0.00 | 0.90±0.00 | 0.90±0.00 | 0.90±0.00 | 0.90±0.00 | 0.89±0.00 | 0.89±0.00 |
| CP (no policy shift) | 0.99±0.00 | 0.94±0.00 | 0.92±0.00 | 0.90±0.00 | 0.88±0.00 | 0.86±0.00 | 0.81±0.00 |

### Table 13: Satimage dataset results

|  | $\Delta_\epsilon = -0.45$ | $\Delta_\epsilon = -0.2$ | $\Delta_\epsilon = -0.1$ | $\Delta_\epsilon = 0.0$ | $\Delta_\epsilon = 0.1$ | $\Delta_\epsilon = 0.2$ | $\Delta_\epsilon = 0.45$ |
|---|---|---|---|---|---|---|---|
| COPP (Ours) | 0.92±0.00 | 0.91±0.00 | 0.91±0.00 | 0.91±0.00 | 0.91±0.00 | 0.91±0.00 | 0.91±0.00 |
| WIS | 1.00±0.00 | 1.00±0.00 | 1.00±0.00 | 1.00±0.00 | 1.00±0.00 | 1.00±0.00 | 1.00±0.00 |
| SBA | 0.98±0.00 | 1.00±0.00 | 1.00±0.00 | 1.00±0.00 | 1.00±0.00 | 1.00±0.00 | 0.99±0.00 |
| COPP (GT behav policy) | 0.90±0.00 | 0.90±0.00 | 0.90±0.00 | 0.90±0.00 | 0.90±0.00 | 0.90±0.00 | 0.89±0.00 |
| CP (no policy shift) | 0.97±0.00 | 0.93±0.00 | 0.92±0.00 | 0.90±0.00 | 0.88±0.00 | 0.87±0.00 | 0.83±0.00 |

# E How the miscoverage depends on $\hat{P}(y \mid x, a)$

**Proposition E.1.** *Let*

$$\tilde{w}(x, y) := \frac{\int \hat{P}(y \mid x, a)\pi^*(a \mid x)\mathrm{d}a}{\int \hat{P}(y \mid x, a)\pi^b(a \mid x)\mathrm{d}a}.$$

*Assume that $\hat{P}(y \mid x, a)/P(y \mid x, a) \in [1/\Gamma, \Gamma]$ for some $\Gamma \geq 1$. Then,*

$$\Delta_w := \tfrac{1}{2}\mathbb{E}_{(X,Y)\sim P^{\pi^b}_{X,Y}} \mid \tilde{w}(X, Y) - w(X, Y) \mid \leq \Gamma^2 - 1.$$

*Proof.* In this proof, we investigate the error of the weights as a function of the error in $\hat{P}(y \mid x, a)$. Therefore, to isolate this effect we ignore the Monte Carlo error, and assume known behavioural policy $\pi^b$.

Under the assumption above, we have that

$$\frac{1/\Gamma \int P(y \mid x, a)\pi^*(a \mid x)\mathrm{d}a}{\Gamma \int P(y \mid x, a)\pi^b(a \mid x)\mathrm{d}a} \leq \tilde{w}(x, y) \leq \frac{\Gamma \int P(y \mid x, a)\pi^*(a \mid x)\mathrm{d}a}{1/\Gamma \int P(y \mid x, a)\pi^b(a \mid x)\mathrm{d}a}.$$

$$\implies \frac{1}{\Gamma^2}w(x, y) \leq \tilde{w}(x, y) \leq \Gamma^2 w(x, y)$$

This means that,

$$\left(\frac{1}{\Gamma^2} - 1\right) w(x, y) \leq \tilde{w}(x, y) - w(x, y) \leq (\Gamma^2 - 1)w(x, y)$$

So,

$$\mid \tilde{w}(x, y) - w(x, y) \mid \leq (\Gamma^2 - 1)w(x, y)$$

And therefore,

$$\mathbb{E}_{(X,Y)\sim P^{\pi^b}_{X,Y}} \mid \tilde{w}(X, Y) - w(X, Y) \mid \leq (\Gamma^2 - 1)\mathbb{E}_{(X,Y)\sim P^{\pi^b}_{X,Y}} [w(X, Y)] = \Gamma^2 - 1$$

$\square$