# OpenReview forum: "Conformal Off-Policy Prediction in Contextual Bandits"
_NeurIPS.cc/2022/Conference — NeurIPS 2022 Accept_

### Official Review · Reviewer_rDgw · 2022-07-09

**Rating:** 7
**Confidence:** 4
**Soundness:** 4 excellent
**Presentation:** 3 good
**Contribution:** 4 excellent

**Summary:**

This paper studies off-policy value predictions at particular X with either marginal or conditional coverage guarantees.  Unlike prior art, this paper preserves the context distribution and models the shift in the outcome distribution via the shift in the action distribution.   The methodology thus applies to stochastic policies and in infinite action spaces (with some caveats).   For marginal guarantees, the methodology works with either a known historical policy or estimated propensity scores.  As is typical in conformal prediction, conditional coverage guarantees require restrictive additional assumptions.

**Questions:**

This paper repeatedly states that known weights are not typical, which contradicts my extensive personal experience with large scale recommendation systems with known randomization.  In causal modeling with observational data propensities are often estimated, but the paper emphasizes the contextual bandit application area.   The estimated weight results are appreciated; but for instance the paper misses a chance to present a simplified corollary of proposition 4.3 for the case when w(x,y) is known.

The moment condition in proposition 4.2 is presented without comment.  Some discussion about how this affects the estimation of \hat{w} would be appreciated (e.g., suggest a regularizer; or comment upon the moment properties of equation (7)).

**Limitations:**

This reviewer is satisfied.

**Strengths And Weaknesses:**

This paper is in the happy position of (essentially) being the first proposed solution to a problem of clear practical importance.

This suffers from the page limit.  Personally, I would elide more of the requisite but uninteresting experimental details to the appendix in order to expand the discussion around proposition 4.3 in the case of known weights.  Alternatively, section 2.2 is ultimately unnecessary: yes prior art used a different covariate shift model, but you could reduce this to a sentence or two in the related work and perhaps a reference to the appendix, and get back some valuable space.

---

> ### Author Response · Authors · 2022-08-02
> **Response to Reviewer rDgw**
>
> We thank the reviewer for the time to review our paper. We appreciate the reviewer's enthusiasm about our work being "the first proposed solution to a problem of clear practical importance". We also thank the reviewer for the valuable suggestions about the presentation for the final version of this paper.
>
> Next we clarify some of the points raised by the reviewer:
>
> >The estimated weight results are appreciated; but for instance the paper misses a chance to present a simplified corollary of proposition 4.3 for the case when w(x,y) is known.
> >
>
> We have tried to keep the theory in our work as general as possible, so it also holds for the case when the weights are known exactly, i.e., in this case, assumption 1 in Proposition 4.3 is known to be true. Additionally, in our ablation study (see Table 1), we have also conducted experiments with known weights (denoted as **COPP (GT weights Ours)** in Table 1 of the paper). Nonetheless, we appreciate this comment and will incorporate this in the final version of the paper.
>
> >The moment condition in proposition 4.2 is presented without comment. Some discussion about how this affects the estimation of $\hat{w}$ would be appreciated (e.g., suggest a regularizer; or comment upon the moment properties of equation (7)).
> >
>
> Thanks for pointing this out. Firstly, the above mentioned moment condition has also been used previously in Lei and Candes 2020 Proposition 1. After further inspection of the proof, we note that in fact, we can replace $r \geq 2$ with $r = 2$. This then translates the moment condition into a finite variance condition. In particular, we can qualitatively interpret this condition as saying that (1) there needs to be sufficient overlap between target and behaviour policy; and (2) estimation of both numerator and denominator in Eq. 7 needs to be sufficiently accurate. Otherwise, if there is too much mass in the tails of $\hat{w}(x, y)$, we may end up computing the quantiles of a degenerate distribution i.e. dirac delta on one score (Eq. 6).
>
> In practice, we could enforce the condition by clipping the value of $\hat{w}$ to ensure it is always finite. This may be a useful strategy for numerical stability of our algorithm more generally, and would lead to minimal changes in practice if the true values of $w$ live between the clipped values. Note however, that we may lose consistency if $w$ lives outside these.
>
> We again thank the reviewer for their time and comments.
>
> [Lei and Candes 2020 "Conformal Inference of Counterfactuals and Individual Treatment Effects"]

---

> > ### Comment · Reviewer_rDgw · 2022-08-08
> > **Acknowledged**
> >
> > I've read your response.
> >
> > > This then translates the moment condition into a finite variance condition.
> >
> > Yeah this is fine, just say something about it.  Finite variance is reasonable enough for now.   Any bounded importance weight will have variance <= range because $E[w]=1$, but in addition some unbounded importance weights still have finite variance.

---

> > > ### Author Response · Authors · 2022-08-08
> > > **Reviewer rDgw**
> > >
> > > We thank the reviewer very much for their response - we agree, and will make the changes in the final version of our paper as suggested.

---

### Official Review · Reviewer_cY8Q · 2022-07-10

**Rating:** 5
**Confidence:** 5
**Soundness:** 3 good
**Presentation:** 3 good
**Contribution:** 3 good

**Summary:**

This paper studies conformal prediction for the (mixed) outcome in contextual bandits under a target policy, using data from a different behavior policy. To tackle the shift from observations to targets, the authors provide the explicit formula for the distributional shift, then propose to use density estimation as well as a Monte Carlo sampling method to compute the conformal prediction intervals. The authors also build on existing methods to show the marginal coverage and asymptotic conditional coverage of the proposed methods. Finally, extensive experiments show the performance of the proposed method.

**Questions:**

1. How does the miscoverage depend on the estimation error of $\hat{P}(y|x,a)$ instead of the hard-to-interpret $\hat{w}$?

2. The density estimation of $\hat{P}(y | x,a)$ is known to be hard because of data scarcity (more difficult than propensity score estimation, at least). It would be helpful to justify this approach by referring to some classical literature on estimation performance under appropriate conditions.

3. Please provide more explanation for comparing to SBA in your experiments.

**Limitations:**

The authors have addressed the limitations and potential negative societal impact of their work.

**Strengths And Weaknesses:**

Originality:

This paper studies a new problem. The techniques contain some novel points but somehow heavily rely on established results. The approach and the results are not too surprising when reading the paper.


-----
Significance:

This paper studies a (moderately new) problem of quantifying the uncertainty in outcomes from contextual bandits. Instead of predicting counterfactuals, the authors consider the mixed overall outcome. This adds to the application of conformal inference, but I am not sure whether this is indeed interesting.


-----
Quality:

The overall quality of this paper is good. The presentation is clear; the theoretical results are easy to understand with nice interpretation and discussions; the comparison to the literature is detailed and comprehensive; the conclusions from experiments are clear.

Besides the strengths, I do have a few concerns.

1. Theories involving weight function estimation.

It is unclear to me how it depends on the Monte Carlo approximation with $h$ samples. Although Proposition 4.2 provides an error bound that is quite similar to Lei and Candes (2021), I am a bit concerned that it is not the most appropriate for this place. Instead, it might be more informative if the error bound could be in terms of $\hat{P}(y | x, a)$ because this quantity is estimated, while $\hat{w}$ is built on top of it.

2. Estimation of $\hat{P}(y | x,a)$.

The density estimation of $\hat{P}(y | x,a)$ is known to be hard because of data scarcity (more difficult than propensity score estimation, at least). It would be helpful to justify this approach by referring to some classical literature on estimation performance under appropriate conditions.


3. Comparison to SBA.

I am not convinced why the authors compare their method to SBA. It is not a conformal-based algorithm and not studied in the literature.

-----
Clarity:

The presentation of this paper is clear and easy to follow. The results are interpreted clearly from time to time. One concern is for the weight function estimation, as stated above.

---

> ### Author Response · Authors · 2022-08-02
> **Response to Reviewer cY8Q**
>
> We thank the reviewer for their time and appreciate the reviewer's comments on the novelty and presentation of the paper.
> We address the points raised by the reviewer as follows:
>
> >How does the miscoverage depend on the estimation error of $\hat{P}(y|x, a)$ instead of the hard-to-interpret $\hat{w}$?
> >
>
> We thank the reviewer for this comment.
> We have included a proof (in the updated appendix E) that bounds the errors in $\hat{w}(x, y)$ using the error in $\hat{P}(y\mid x, a)$, under certain assumptions on $\hat{P}(y\mid x, a)$.
> We emphasise that the conditions in this proof are sufficient but not necessary.
> In particular we can have situations where $\hat{P}(y\mid x, a)$ may be arbitrarily different from $P(y\mid x, a)$, even though the estimated weights $\hat{w}(x, y)$ are exactly accurate.
> We have given such an example in appendix B.4.1.
>
> Additionally, we would like to note that the estimation of $\hat{w}(x, y)$ does not necessarily have to involve $\hat{P}(y\mid x, a)$ as we outline an alternative methodology of learning weights directly in appendix B.4.2. This means that we are not forced to learn $\hat{P}(y\mid x, a)$ for our algorithm COPP to work.
>
> We are currently exploring extensions using the alternative methods in appendix B.4.2, however, this is outside the scope of this paper.
>
> >The density estimation of $\hat{P}(y\mid x, a)$
> is known to be hard because of data scarcity (more difficult than propensity score estimation, at least). It would be helpful to justify this approach by referring to some classical literature on estimation performance under appropriate conditions.
> >
>
> While we agree that continuous density estimation is indeed a hard task in general, we would like to note that in most recommendation system settings, the data may not be scarce and the outcome is categorical. This means that $\hat{P}(y\mid x, a)$ estimation can be achieved by training a classifier for $(x, a) \rightarrow y$, which in general is much easier than continuous density estimation.
>
> Additionally, we highlight in line 145 of our paper, that in theory, we are not required to estimate $\hat{P}(y\mid x, a)$. We present an alternative methodology of learning the weights $\hat{w}(x, y)$ directly in appendix B.4.2. that do not depend on $\hat{P}(y\mid x, a)$, which can be leveraged in continuous/discrete cases.
>
> However, in our paper, we used the naive approach of computing the weights, i.e. through $\hat{P}(y\mid x, a)$, as it worked very well empirically for our experiments. We leave the alternative methodology for future work.
>
> >Please provide more explanation for comparing to SBA in your experiments
> >
>
> As we noted in our paper [line 138], we mainly use SBA as a direct comparison against COPP to ablate the result of using Conformal Prediction, as opposed to directly using $\hat{P}(y\mid x, a)$ to construct the intervals. The latter method corresponds to SBA. This ablation was performed to ensure that CP is, in fact, needed to achieve the required coverage guarantees. In Table 1, we have provided other baselines that are more ``conformal'' in nature, against which COPP still performs better.
>
> We will clarify in the final version that we do not consider SBA a standard baseline, however, given our methodology, it is a natural comparison to make to answer the question `why not construct the intervals using $\hat{P}(y\mid x, a)$ directly?'.
>
> We thank the reviewer for this comment and we will add additional details on why we have added SBA in our experiments in the final version of the paper.
>
> > This adds to the application of conformal inference, but I am not sure whether this is indeed interesting.
> >
>
> We respectfully disagree with the reviewer, as we think that COPP is interesting both, from an application and a methodological perspective. As pointed out by reviewer 3, this work is `the first proposed solution to a problem of clear practical importance' in particular recommendation systems. Other works have explored uncertainty quantification in off-policy evaluation (Dai et al 2020), however, to the best of our knowledge COPP is the first algorithm tackling uncertainty quantification in off-policy outcomes, that provides finite sample coverage guarantees.
>
> In recommendation systems, knowing the likely range of outcomes can be useful as it reduces the cost of testing new policies (for example, through A/B testing). From a methodological perspective, we believe that the reformulation of policy shift as a conditional shift problem in conformal setting is a novel contribution to the conformal literature as it has not been explored before. Previous works only considered covariate and label shifts, but not conditional shift.
>
> We hope that we were able to address all the reviewer's questions in the above and hope that the reviewer would consider increasing their score.
>
> [Dai et al 2021 "CoinDICE: Off-Policy Confidence Interval Estimation"]

---

### Official Review · Reviewer_E2XW · 2022-07-10

**Rating:** 5
**Confidence:** 3
**Soundness:** 3 good
**Presentation:** 3 good
**Contribution:** 2 fair

**Summary:**

The paper studies how to construct predictive intervals from a contextual bandit policy such that the true labels are within the predictive intervals with a user specific probability, using some bandit feedback data, which they call the calibration data.

The paper identifies that the data distributions are different under the logging policy and the target policy in contextual bandits, which impedes using conventional conformal prediction methods to provide predictive intervals with coverage guarantees. So they adopt the conformal prediction method with covariate shifts [13] to this problem to provide coverage guarantees despite the data distribution shift.

They provide marginal coverage (adapted from prior works) and conditional coverage analysis (with strong assumptions) of the method, and demonstrate its effectiveness in empirical evaluation.

**Questions:**

See weaknesses.

**Limitations:**

I did not find limitations and potential negative societal impact.

**Strengths And Weaknesses:**

The paper is well-written and easy to follow.

They provide coverage analysis of the proposed method.

The empirical evaluation demonstrates the effectiveness of the proposed method.

Weaknesses:

1. The method does not provide any optimality guarantee. The method does provide coverage guarantee (i.e. the prediction set contains the true label with high probability). But it does not have any guarantee on the size of the prediction sets. I can easily construct prediction sets with perfect coverage guarantee, i.e. the set of all labels.

2. The paper does not seem to provide new technical contributions. The proofs seem to be directly adapted from prior works.

3. I am also a bit confused about whether the guarantees are desirable. In proposition 4.1, the randomness is in both the calibration data and the one test data point, right? Does this mean that for each single test data, we need n calibration data to construct the prediction sets to provide the coverage guarantee? Would it be possible to get this kind of guarantee instead: with high probability in the calibration data, the probability that the true label is in the prediction set is greater than a user-specified level 1-\alpha? If this is true, it then makes sense to construct a single set prediction function using calibration data, and then apply it to a set of test data.

---

> ### Author Response · Authors · 2022-08-02
> **Response to Reviewer E2XW -- Part I**
>
> We thank the reviewer for their time spent reviewing our manuscript.
> We are pleased to hear that the reviewer found our paper easy to read and follow.
> We also appreciate the reviewer's acknowledgement of our extensive coverage analysis and empirical evaluation of our proposed method COPP.
> We address the questions raised by the reviewer below:
> > The method does not provide any optimality guarantee. The method does provide coverage guarantee (i.e. the prediction set contains the true label with high probability). But it does not have any guarantee on the size of the prediction sets. I can easily construct prediction sets with perfect coverage guarantee, i.e. the set of all labels.
> >
> We note that in addition to the lower bound on coverage, our Proposition 4.2 also includes an upper bound for our proposed method. This upper bound ensures that the sets constructed using COPP are not too conservative in the marginal sense, i.e., the sets have close to $1-\alpha$ marginal coverage as the number of calibration points grows large.
> This upper bound on the guarantee hence rules out the trivial coverage set containing all labels suggested by the reviewer, as the coverage in that case would be always equal to 1.
>
> We additionally point out that our experiments show that in practice COPP is able to produce nontrivial intervals while maintaining desired the coverage lower bound guarantee - see e.g. Figure 1b where the COPP intervals are closer to the oracle conditional intervals than any other baseline.
>
> > The paper does not seem to provide new technical contributions. The proofs seem to be directly adapted from prior works.
> >
> We respectfully disagree that our paper does not provide new technical contributions.
> To the best of our knowledge, our paper is the first to consider uncertainty quantification in off-policy outcomes, as opposed to the expected value of the policy.
> Additionally, we believe that formulating the off-policy prediction as conditional shift in the conformal setting is novel, and distinct from e.g. previous approaches such as Tibshirani et al. (2018) or Lei & Candes (2020) who instead formulate the problem as covariate shift, or Podkopaev et al. (2021) who consider label shift.
>
> As a result of our alternative approach, our proposed method COPP is also applicable to continuous action spaces and stochastic target policies.
> In addition, our approach naturally leads to other interesting possibilities such as the novel methodology of learning weights $\hat{w}(x, y)$ outlined in Appendix B.4.2.
> This is, to the best of our knowledge, an original algorithm for regressing to the conditional ratios directly, and we are currently exploring this in our follow-up works.
>
> Finally, while some of our theoretical results (which we note e.g. in lines 151 \& 160) are indeed extensions of prior work which we include for completeness, our main result of Proposition 4.3 is not. The motivation for Proposition 4.3 was inspired by the Doubly Robust results in (Lei and Candes 2020), however the proposition itself is original and the proof in appendix A.3 requires various nontrivial and novel ideas that are specific to our context.

---

> > ### Comment · Reviewer_E2XW · 2022-08-08
> > **Thanks for the response**
> >
> > The authors addressed my first two concerns. I still have concern about the third weakness. Is the empirical evaluation consistent with the theoretical guarantees? Do you sample n validation data points for each test data point? If yes, I do not think this is realistic. If not, it is inconsistent with your theoretical guarantees.

---

> > > ### Author Response · Authors · 2022-08-08
> > > **Reply to Reviewer E2XW**
> > >
> > > We thank the reviewer very much for their reply.
> > >
> > > Our experiments have the following structure: for each method reported in tables, we use 10 runs, in each of which we sample one calibration dataset $D_{cal}$ of size $n$, which we subsequently use to construct the conformal predictor $C$ for all test points in that run.
> > >
> > > By using a large number of i.i.d. test points, we are able to obtain a reliable approximation of the coverage conditional on the calibration set, i.e. $P_{X_{test}, Y_{test}}(Y_{test} \in C(X_{test}) \mid D_{cal})$.
> > > We then compute the mean and standard error of these conditional probabilities across the 10 runs.
> > > Since each of these has a separate i.i.d. calibration set $D_{cal}$, we thereby gain an approximation of
> > > $$
> > >     P_{D_{cal}, X_{test}, Y_{test}}(Y_{test} \in C(X_{test})) = E_{D_{cal}}[P_{test}(Y_{test} \in C(X_{test}) \mid D_{cal})].
> > > $$
> > >
> > > Our experiments are therefore correctly set up to verify the theoretical guarantees that we prove, and consistently show that we obtain the required coverage of $1 - \alpha$ on average over $D_{cal}$.
> > >
> > > We note that the coverage guarantee considered by conformal prediction in general holds only on average, i.e. it is not the case that $P_{X_{test}, Y_{test}}(X_{test} \in C(X_{test}) \mid D_{cal}) \geq 1 - \alpha$ for *every* calibration set $D_{cal}$.
> > > However, as shown in our results Tables 1a and 2, the standard errors obtained in our estimation of $P_{X_{test}, Y_{test}}(Y_{test} \in C(X_{test}))$ by averaging $P_{X_{test}, Y_{test}}(Y_{test} \in C(X_{test}) \mid D_{cal})$ across multiple calibration sets are very small ($\leq 0.02$).
> > >
> > > This shows empirically that the distribution of this coverage probability $P_{test}(Y_{test} \in C(X_{test}) \mid D_{cal})$ is highly peaked around its nominal coverage, so that in fact the probability that any $D_{cal}$ gives rise to the correct coverage is high.
> > > We do not make any theoretical claims that this will always hold, although as we note in our earlier reply there is an increasing body of literature that does explore this alternative type of guarantee in this context as well that may be relevant to consider in future work.
> > > Nevertheless, the marginal guarantee that we consider theoretically remains of significant interest in the conformal prediction literature [Angelopoulos et al. 2022] since it provides a means of constructing $C$ with good predictive performance on average, which has proven useful across many applications.
> > >
> > > We also emphasise that we only used multiple calibration sets to evaluate our methodology for the purpose of this paper.
> > > At deployment time the same predictor $C$ produced by a fixed i.i.d. calibration set $D_{cal}$ can be reused multiple times to make predictions across a batch of i.i.d. test points, and the predictor will obtain the same marginal coverage guarantee across each individual test point (and the calibration set).
> > >
> > > Explicitly, suppose we have i.i.d. test points $(X_{test}^{(i)}, Y_{test}^{(i)})$ (where as usual the $Y_{test}^{(i)}$ values are unknown).
> > > Since these points are independent of each other and $D_{cal}$ (and hence independent of $C$), it follows that
> > >
> > > $$
> > > P_{(X_{test}^{(i)}, Y_{test}^{(i)})} (Y_{test}^{(1)} \in C(X_{test}^{(1)}), \ldots, Y_{test}^{(k)} \in C(X_{test}^{(k)}) | D_{cal}) = $$
> > > $$
> > > P_{X_{test}^{(1)}, Y_{test}^{(1)}}(Y_{test}^{(1)} \in C(X_{test}^{(1)}) | D_{cal}) \ldots P_{X_{test}^{(k)}, Y_{test}^{(k)}}(Y_{test}^{(k)} \in C(X_{test}^{(k)}) | D_{cal})
> > > $$
> > >
> > > In other words, given any possible calibration dataset, the coverage probabilities are i.i.d., so the predictor will produce the same performance across each test point.
> > > Moreover, marginally, each has the desired coverage that we proved, i.e. for any $i \in \{1, \ldots, k\}$
> > > $$
> > >     P_{X_{test}^{(i)}, Y_{test}^{(i)}}(X_{test}^{(i)} \in C(X_{test}^{(i)})) = E_{D_{cal}}[P_{X_{test}^{(i)}, Y_{test}^{(i)}}(X_{test}^{(i)} \in C(X_{test}^{(i)}) \mid D_{cal})] \geq 1 - \alpha.
> > > $$
> > >
> > > We hope that this will mitigate any concerns that our proposed guarantee and workflow (which we emphasise again are standard for conformal prediction [Angelopoulos et al. 2022]) are not realistic for use in practice.
> > >
> > > We thank the reviewer again for their time spent reviewing our paper.
> > > We would be very happy to clarify any further concerns or questions, or otherwise hope that the reviewer will consider increasing their score.
> > >
> > > [Angelopoulos et al. 2022 "A gentle introduction to conformal prediction and distribution-free uncertainty quantification"]

---

> ### Author Response · Authors · 2022-08-02
> **Response to Reviewer E2XW -- Part II**
>
> > I am also a bit confused about whether the guarantees are desirable. In proposition 4.1, the randomness is in both the calibration data and the one test data point, right? Does this mean that for each single test data, we need n calibration data to construct the prediction sets to provide the coverage guarantee? Would it be possible to get this kind of guarantee instead: with high probability in the calibration data, the probability that the true label is in the prediction set is greater than a user-specified level $1-\alpha$? If this is true, it then makes sense to construct a single set prediction function using calibration data, and then apply it to a set of test data.
> >
>
> We thank the reviewer for raising this interesting point.
> We have concentrated on obtaining a marginal coverage guarantee given its prominence in the standard conformal literature, which has proven to be of significant interest in many contexts (see Angelopoulos et al. 2022 for review).
> However, we agree that it would indeed be worth considering alternative coverage guarantees as suggested.
> We note that some recent work in distribution-free uncertainty quantification (called "Risk-Controlling prediction" (Bates et al. 2021) and "Learn then Test" (Angelopoulos et al. 2021)) already exists that does control for the calibration dataset, and which can be used for example to achieve the following guarantee:
> $$
> P_{cal} [P_{test}(Y_{test} \in \hat{C}(X_{test})\mid \mathcal{D}_{cal} ) \geq 1-\alpha] \geq 1- \delta
> $$
> which corresponds to the reviewer's criterion.
>
> It has also been shown in (Vovk 2012, Angelopoulos et al 2022, Section 3.1) that conditioned on calibration data, the standard conformal sets achieve coverage which is `close' to the required coverage with high probability.
> It may also be possible to extend this result to apply in the case of distribution shift: this is yet to be explored but is an interesting future direction for research in the Conformal literature.
>
> We believe it is outside the scope of our current paper to consider how to apply this literature to our context but believe that this would constitute important future work, and will mention this and cite these papers accordingly in the next version of our paper.
>
> We hope that the above has addressed the questions raised by the reviewer adequately, and that the reviewer will consider raising the score.
>
> [Vovk 2012 "Conditional validity of inductive conformal predictors”]
>
> [Bates et al 2021 "Distribution-Free, Risk-Controlling Prediction Sets"]
>
> [Angelopoulos et al. 2022 "A gentle introduction to conformal prediction and distribution-free uncertainty quantification"]
>
> [Angelopoulos et al. 2021 "Learn then Test: Calibrating Predictive Algorithms to Achieve Risk Control"]
>
> [Lei and Candes 2020 "Conformal Inference of Counterfactuals and Individual Treatment Effects"]
>
> [Tibshirani et al 2020 "Conformal Prediction Under Covariate Shift"]
>
> [Podkopaev et al 2021 "Distribution-free uncertainty quantification for classification under label shift"]

---

### Meta-Review · Area_Chair_Kq6a · 2022-08-24

**Recommendation:** Accept
**Confidence:** Less certain

**Metareview:**

The paper develops conformal prediction tools for OPE under the contextual bandit model. The reviewers find the problem of clear importance and the quality of the paper sufficiently high.  All reviewers are in favor of acceptance. The authors are encouraged to incorporate the reviewers' feedback and have a clear discussion of the technical novelty in the final version.

**Award:**

No

---

### Decision · Program_Chairs · 2022-09-14

Accept